# Self-assembled organic nanomedicine enables ultrastable photo-to-heat converting theranostics in the second near-infrared biowindow

Huijing Xiang[1,2,3], Lingzhi Zhao[2], Luodan Yu[1,3], Hongzhong Chen[2], Chenyang Wei[3], Yu Chen [1,3✉] & Yanli Zhao [2✉]

Development of organic theranostic agents that are active in the second near-infrared (NIR-II, 1000–1700 nm) biowindow is of vital significance for treating deep-seated tumors. However, studies on organic NIR-II absorbing agents for photo-to-heat energy-converting theranostics are still rare simply because of tedious synthetic routes to construct extended π systems in the NIR-II region. Herein, we design a convenient strategy to engineer highly stable organic NIR-II absorbing theranostic nanoparticles (Nano-BFF) for effective phototheranostic applications via co-assembling first NIR (NIR-I, 650–1000 nm) absorbing boron difluoride formazanate (BFF) dye with a biocompatible polymer, endowing the Nano-BFF with remarkable theranostic performance in the NIR-II region. In vitro and in vivo investigations validate that Nano-BFF can serve as an efficient theranostic agent to achieve photoacoustic imaging guided deep-tissue photonic hyperthermia in the NIR-II biowindow, achieving dramatic inhibition toward orthotopic hepatocellular carcinoma. This work thus provides an insight into the exploration of versatile organic NIR-II absorbing nanoparticles toward future practical applications.

[1] School of Life Sciences, Shanghai University, Shanghai 200444, P. R. China. [2] Division of Chemistry and Biological Chemistry, School of Physical and Mathematical Sciences, Nanyang Technological University, 21 Nanyang Link, Singapore 637371, Singapore. [3] State Key Laboratory of High Performance Ceramics and Superfine Microstructure, Shanghai Institute of Ceramics, Chinese Academy of Sciences, Shanghai 200050, P. R. China. ✉email: chenyuedu@shu.edu.cn; zhaoyanli@ntu.edu.sg

Photo-to-heat energy-converting theranostics have garnered comprehensive attention as highly promising cancer-treatment strategies for thermal ablation of tumor cells[1–3]. Upon exposure to near-infrared (NIR) laser, NIR-absorbing agents with high photothermal conversion efficiency can produce hyperthermia to promote cancer-cell apoptosis[4–6]. There is an urgent need to extend the excitation wavelength to the second NIR (NIR-II, 1000–1700 nm) region[7–10]. In comparison with the first NIR (NIR-I, 650–1000 nm) biowindow, NIR-II biowindow possesses better tissue-penetration capability owing to smaller amount of light scattering in biological tissues and minimized tissue-background signal[11–14]. Moreover, NIR-II biowindow imparts higher maximum permissible exposure (MPE, for instance, 1 W cm$^{-2}$ for 1000–1100 nm) to NIR-II laser[15–17]. Therefore, the exploration of desirable photothermal agents with high light-to-heat conversion efficiency in the NIR-II biowindow for efficient phototheranostic application is highly important to augment the anticancer efficacy in treating deep-seated tumors[18,19]. However, only a few NIR-II-photoabsorbing nanomaterials, such as gold nanostructures[20,21], MXenes[22,23], copper sulfide nanoparticles[24,25], ammonium tungsten bronze nanoparticles[26,27], and large conjugated polymer nanoparticles[15,28], have been reported for their photothermal performance in vivo.

In comparison with inorganic NIR-II-absorbing nanomaterials, organic photothermal agents present a superior promise for clinical translation in virtue of their intrinsic features including definite chemical structure, excellent reproducibility, high biocompatibility, and desirable biodegradability[29,30]. On the other hand, the exploitation of theranostic applications of NIR-II-absorbing dyes has been hindered by their tedious synthetic protocols, and extraordinarily rare successful paradigms have been validated[31]. Enlightened by the engineering principles found in natural light-harvesting systems, the self-assembly of dye molecules has attracted increasing interest, as the aggregated state may show new photophysical characteristics that are not found in the monomers[32–35]. Numerous biomimetic supramolecular architectures have been assembled from synthetic dyes such as porphyrins[29], merocyanines[33], chlorins[36], naphthalenes[37], and boron dipyrromethenes[38], the aggregation of which could be tailored from the H-type (hypsochromic shift) to the desired J-type (bathochromic shift). Therefore, rational modulation of supramolecular architectures provides significant insights into the construction of dye aggregates with pronounced bathochromic shifts into the NIR-II spectral domain distinct from the absorption of corresponding monomers in the NIR-I region.

Herein, NIR-II absorbing organic nanoparticles (Nano-BFF) were developed by simply self-assembling an ultrastable NIR-I-photoabsorbing boron difluoride formazanate (BFF) dye with an amphiphilic polymer, which could serve as a highly efficient phototheranostic agent in the NIR-II biowindow. On account of strong π–π stacking interaction between dye molecules, the self-assembled aggregates revealed broad absorption in the NIR-II region relative to that of individual dye in the NIR-I region. Importantly, the constructed Nano-BFF exhibited high photothermal stability, excellent photobleaching resistance, strong photoacoustic (PA) imaging signal, and enhanced photothermal conversion in the NIR-II biowindow. It is noteworthy that the Nano-BFF could be used as a superior organic photoabsorbing agent to implement in vivo PA imaging-guided deep-tissue photonic hyperthermia in the NIR-II biowindow, achieving significant tumor elimination in an orthotopic hepatocellular carcinoma (HCC) model.

## Results

**Fabrication and characterization of Nano-BFF.** The readily accessible NIR-I-photoabsorbing BFF complex was adopted as an ideal dye for constructing NIR-II-absorbing nanomedicine, owing to its simple and low-cost synthetic routes as well as attractive characteristics such as unique spectroscopic properties, high absorption coefficient, and intrinsic chemical and photochemical stability. In addition, it shows a great potential in a wide range of applications, covering molecular probes, cellular imaging agents, and photosensitizers[39,40]. The detailed synthetic route of BFF was illustrated in the Supplementary Information (Section S1). The identity and purity of intermediates and final product were fully verified (Supplementary Figs. 1–7). To enable BFF as an in vivo versatile theranostic agent in the NIR-II biowindow, biocompatible Nano-BFF nanoparticles were constructed via assembling BFF dye with amphiphilic biocompatible polymer F127. After the aggregation, Nano-BFF manifested a pronounced bathochromic shift into the NIR-II spectral domain as compared to the absorption of individual BFF dye in the NIR-I region, thus endowing Nano-BFF with a phototheranostic potential in treating deep-seated tumors under NIR-II laser irradiation (Fig. 1).

The morphology of Nano-BFF was confirmed by transmission electron microscopy, revealing monodispersed spherical morphology with the mean size of ~140 nm (Supplementary Fig. 8a). The hydrodynamic diameters of Nano-BFF were determined to be 161, 193, and 166 nm in the medium of $H_2O$, Dulbecco's modified eagle medium, and phosphate buffer solution (PBS) by dynamic light scattering (DLS) measurements, respectively, demonstrating the presence of individual Nano-BFF nanoparticles in aqueous solution (Supplementary Fig. 8b). Notably, Nano-BFF exhibited broad absorption in the NIR-II biowindow ranging from 900 nm to 1200 nm in comparison with that of the BFF dye in the NIR-I region (Supplementary Fig. 8c). In addition, the molar extinction coefficient of Nano-BFF at 808 nm and 1064 nm was calculated to be $3.07 \times 10^5$ and $2.22 \times 10^5\,M^{-1}\,cm^{-1}$ respectively, which was comparable or higher than that of ICG ($2.16 \times 10^5\,M^{-1}\,cm^{-1}$ at 795 nm)[41] and Mito-CCy ($1.55 \times 10^5\,M^{-1}\,cm^{-1}$ at 734 nm)[42], indicating that Nano-BFF is a promising photoabsorbing agent to achieve PA imaging-guided photonic hyperthermia in the NIR-II biowindow (Supplementary Fig. 9).

The photothermal stability and photobleaching/aggressive agent resistance are crucial for PA/photothermal performance in vitro and in vivo. We thus assessed the photostability of BFF, Nano-BFF, clinically used indocyanine green (ICG), and Au nanorods (Au NRs) upon NIR-I or NIR-II laser irradiation. The absorbance of ICG approximately dropped to zero after exposure to repeated NIR-I laser irradiation for 7 min. In addition, the evident decrease and blue-shift could be observed in the characteristic band of Au NRs, owing to the significant aggregation of Au NRs under continuous NIR-I laser irradiation (Supplementary Fig. 10). In contrast, the absorption spectra of BFF and Nano-BFF were nearly unchanged during NIR-I laser irradiation period (Supplementary Fig. 11b–d, f). Especially, NIR-II laser illumination exhibited a negligible effect on the optical property of Nano-BFF (Supplementary Fig. 11e, f). Correspondingly, the colors of BFF and Nano-BFF were identical to their original states after repeated NIR-I or NIR-II laser illumination for 7 min, whereas the color of ICG aqueous solution changed gradually from green to yellow–green under the same conditions (Supplementary Fig. 11a). Furthermore, we investigated the photobleaching resistance of Nano-BFF and ICG aqueous solution via alternative heating and cooling process. The photothermal conversion performance of Nano-BFF showed negligible variation after three cycles of heating/cooling processes, whereas the temperature elevation of ICG solution significantly declined to ~15% ($\Delta T = 5\,°C$) of the original value ($\Delta T = 33\,°C$) after three cycles of irradiation, demonstrating that Nano-BFF had higher resistance to photobleaching in sharp contrast to the

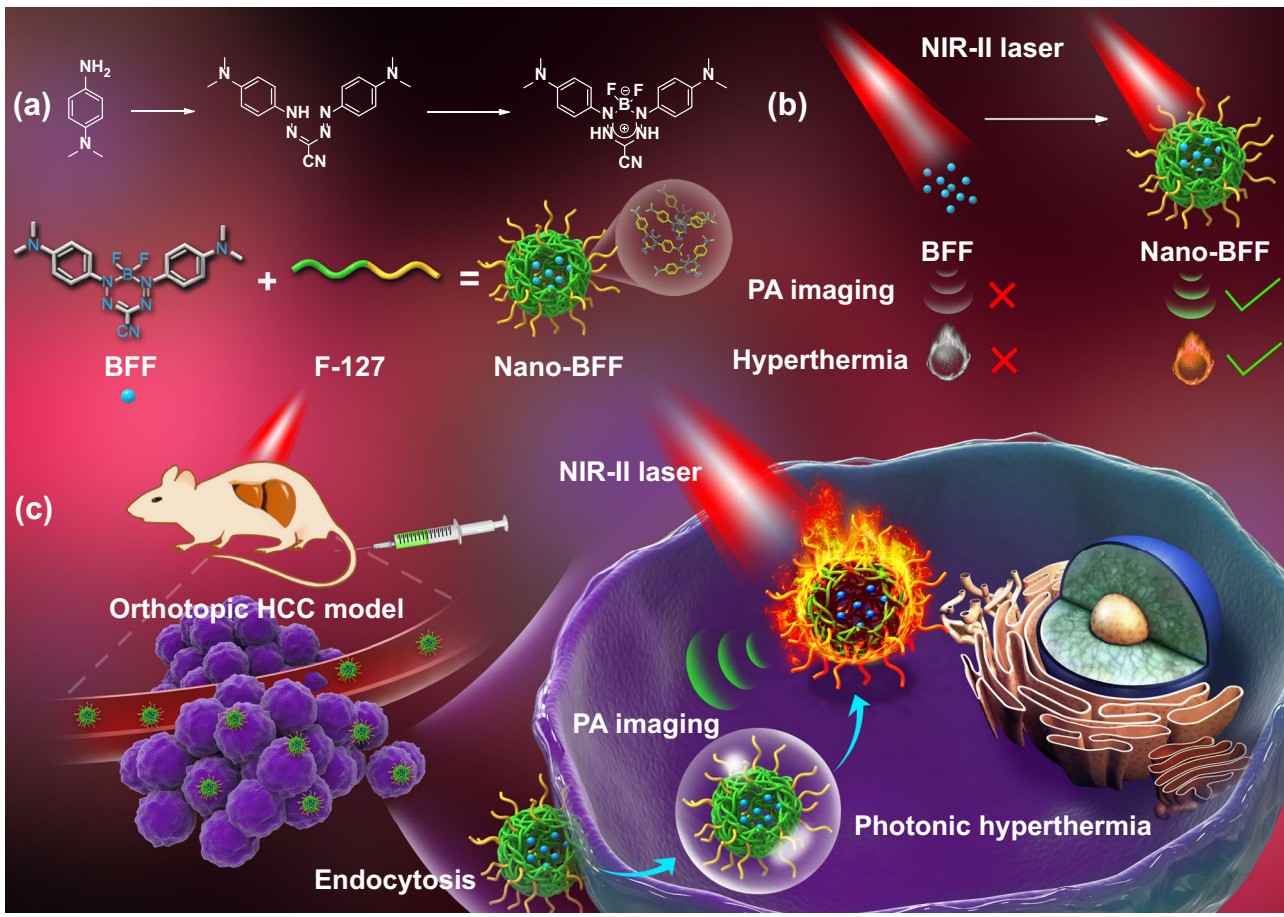

**Fig. 1 Schematic construction of Nano-BFF and its theranostic application. a** Schematic illustration for self-assembled construction of Nano-BFF. **b**, **c** Its specific theranostic application in PA imaging-guided deep-tissue photonic hyperthermia in the NIR-II biowindow.

clinically used ICG (Supplementary Fig. 11g, h). Furthermore, DLS analysis was performed to assess the stability of Nano-BFF under acidic physiological condition. The particle size of Nano-BFF at pH 5.5 and pH 6.5 remained stable without obvious aggregation over 24 h, suggesting high stability of Nano-BFF under acidic tumor microenvironment (Supplementary Fig. 12). To evaluate the physiological stability of Nano-BFF against aggressive agents, we then recorded the absorption spectra of ICG, BFF, and Nano-BFF in the absence and presence of two types of reactive molecules, hydroxyl radical ($\cdot$OH) and hypochlorite ion ($ClO^-$). The absorbance of ICG at 780 nm dropped ~87% and 95% relative to the original values after the addition of $\cdot$OH and $ClO^-$, respectively. In sharp contrast, negligible change was observed in the absorption spectra of BFF and Nano-BFF in the presence of $\cdot$OH and $ClO^-$ (Supplementary Fig. 13a–d), revealing that they had high resistance to oxidation by reactive molecules. Therefore, Nano-BFF could serve as an ultrastable phototheranostic agent for PA imaging-guided disease treatment even in the presence of some reactive molecules.

**In vitro photonic hyperthermia performance of Nano-BFF.** Inspired by the intense absorption of Nano-BFF in both NIR-I and NIR-II biowindow, we exploited the in vitro photothermal conversion performance of Nano-BFF under 808 nm and 1064 nm laser exposure. There was a significant temperature increase when the power density of the laser varied from 0.5 to 1.5 W cm$^{-2}$ (Fig. 2a, e). The temperature of Nano-BFF aqueous solution (200 μg mL$^{-1}$) could rapidly reach to 63 °C and 61 °C under

808 nm (1.5 W cm$^{-2}$) and 1064 nm (1.5 W cm$^{-2}$) laser irradiation for 5 min, respectively (Supplementary Fig. 14a, b). In addition, the temperature of Nano-BFF solution at varied concentrations was monitored by infrared (IR) thermal images, and the temperature elevation was both irradiation duration-dependent and dose-dependent (Fig. 2b, f), whereas negligible temperature increment was detected for deionized water under the same conditions (Supplementary Fig. 14c, d). Subsequently, the photothermal conversion efficiency of Nano-BFF was determined to be 28.6% and 34.3% at 808 nm and 1064 nm, respectively (Fig. 2c, g), which is significantly higher than that of various photothermal agents (e.g., ICG 3.1%[42], IR1048-MZ 20.2%[43], and Au NRs 21%[44]). These findings directly proved that Nano-BFF could efficiently and rapidly convert photo energy into heat under laser irradiation. To evaluate the photothermal stability of Nano-BFF, the temperature elevation profiles of Nano-BFF aqueous solution under six cycles of heating and cooling processes were recorded. The heating behavior of the solution revealed no significant deterioration during each cycle test, indicating that Nano-BFF could be adopted as a durable photo-absorbing agent for photonic cancer hyperthermia in the NIR-II biowindow (Fig. 2d, h).

To assess the deep-tissue-penetration capability of both NIR-I and NIR-II lasers, the power density of 808 nm or 1064 nm laser after penetration into the chicken breast tissue with increasing thickness (2, 4, 6, 8, and 10 mm) was monitored by an optical power meter (Fig. 2i, j). In comparison with 1064 nm laser, the residue power density of 808 nm laser attenuated more rapidly at each tissue depth. For instance, after the penetration through 2 mm of chicken breast tissue, the residue power density for 1064 nm and

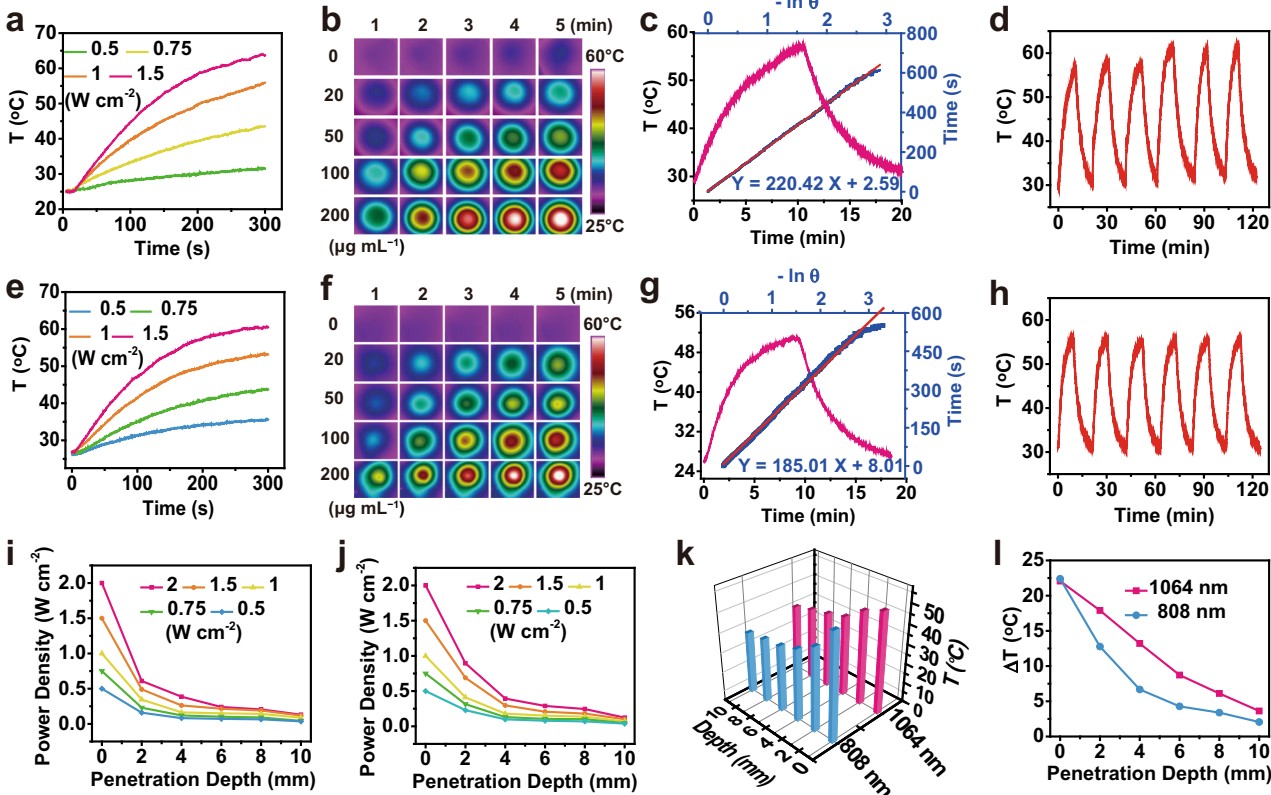

**Fig. 2 Photo-to-heat energy conversion performance of Nano-BFF. a, e** Photothermal conversion curves of Nano-BFF aqueous solution (200 μg mL$^{-1}$) under **a** 808 nm and **e** 1064 nm laser irradiation at varied power densities. **b, f** IR thermal images of Nano-BFF aqueous solution at varied concentrations under **b** 808 nm (1.5 W cm$^{-2}$) and **f** 1064 nm (1.5 W cm$^{-2}$) laser irradiation. **c, g** Photothermal effect of Nano-BFF aqueous solution after irradiation with **c** 808 nm and **g** 1064 nm lasers and linear fitting curves between time and −lnθ acquired from the cooling process. **d, h** On−off curves of Nano-BFF aqueous solution (100 μg mL$^{-1}$) upon **d** 808 nm (1.5 W cm$^{-2}$) and **h** 1064 nm (1.5 W cm$^{-2}$) laser irradiation. **i, j** Power density of **i** 808 nm and **j** 1064 nm lasers after penetration through tissues with different thicknesses. **k** Real-time temperature, and **l** temperature variations of Nano-BFF aqueous solution upon the laser irradiation after the penetration into tissues with increasing skin thickness.

808 nm lasers was determined to be 0.9 W cm$^{-2}$ and 0.6 W cm$^{-2}$ under the same original power density, respectively. This difference was contributed to better transmittance capability and higher MPE of 1064 nm laser in comparison with that of 808 nm laser[45–47]. To further verify higher deep-tissue photothermal heating performance of NIR-II laser, Nano-BFF solution was irradiated by 1064 nm or 808 nm laser under varied tissue thicknesses (Fig. 2k, l). The temperature increment of Nano-BFF solution irradiated by 1064 nm with the penetration depth of 2 mm reached 17.9 °C, which was higher than that of 12.8 °C for 808 nm laser under the same conditions (Supplementary Fig. 14e, f), attributing to more prominent deep-tissue-penetration capability together with higher MPE limit of 1064 nm laser as compared with that of 808 nm laser.

**In vitro photonic hyperthermia of Nano-BFF against cancer cells.** Given the desirable photothermal conversion performance of Nano-BFF, we then assessed its in vitro photothermal therapy (PTT) efficacy against cancer cells (Fig. 3a). The intracellular uptake of Nano-BFF at various time points was evaluated by confocal laser scanning microscopy (CLSM) using cyanine 3 (Cy3)-labeled Nano-BFF (Fig. 3b). The CLSM images confirmed that Nano-BFF could be efficiently internalized into 4T1 cells, as observed by the red fluorescence derived from Cy3-labeled Nano-BFF after 2 h of incubation. The intensity of Cy3 fluorescence originated from Nano-BFF kept increasing up to 12 h, further verifying that the degree of intracellular uptake augmented with

the prolonged incubation time. Subsequently, the dark cytotoxicity of Nano-BFF was studied by cell counting kit-8 (CCK-8) assay after the incubation with increasing doses of Nano-BFF for 24 and 48 h, respectively. Nano-BFF did not cause a marked decrease in the cell survival rate after incubation for 48 h, even at the maximum dose of 200 μg mL$^{-1}$, proving its high biocompatibility toward 4T1 and MCF-7 cells (Fig. 3c and Supplementary Fig. 15a). The in vitro PTT efficiency of Nano-BFF was investigated by using 4T1 cells as the model cell line. Laser irradiation or Nano-BFF incubation at defined conditions induced a negligible effect on cell survival rate (Supplementary Fig. 15b). Nano-BFF-induced PTT effect highly relies on its dosage. For instance, when the dose of Nano-BFF was increased to 200 μg mL$^{-1}$, ~90% of the 4T1 cells were thermally ablated under both NIR-I and NIR-II laser irradiation, suggesting prominent photothermal conversion performance of Nano-BFF in promoting tumor cell death (Fig. 3d, e). Meanwhile, the PTT efficacy of Nano-BFF was also evaluated by calcein acetoxymethyl ester/propidium iodide (calcein-AM/PI) assay. For the treatment groups of PBS, Nano-BFF, NIR-I laser, and NIR-II laser, almost all of the cells showed intense green fluorescence, indicating negligible damage effect of Nano-BFF administration and laser illumination on cancer cells under the defined conditions. In contrast, bright red fluorescence was evidently observed in the treatment groups of Nano-BFF + NIR-I laser and Nano-BFF + NIR-II laser, proving excellent photothermal capability of Nano-BFF in both NIR-I and NIR-II biowindow (Fig. 3f).

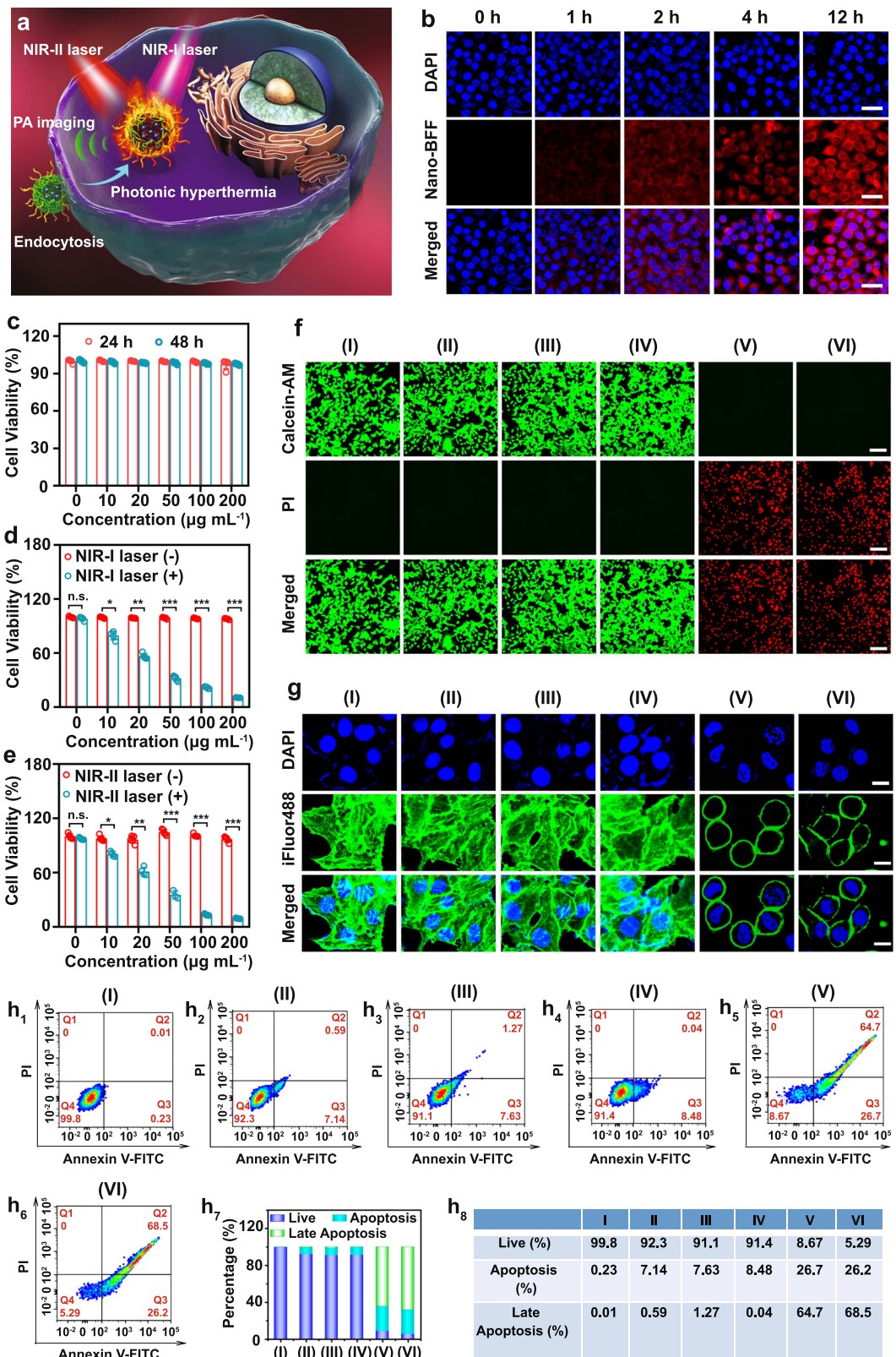

**Fig. 3 Intracellular uptake and in vitro photonic hyperthermia of Nano-BFF. a** Scheme of in vitro PTT treatment by Nano-BFF. **b** Confocal images of 4T1 cells administrated with Cy3-labeled Nano-BFF at varied time points (scale bars: 50 μm). **c** Cell survival rates of 4T1 cells administrated with Nano-BFF at elevated doses for 24 and 48 h ($n = 5$ biologically independent samples). **d**, **e** Cell survival rates of 4T1 cells treated with various concentrations of Nano-BFF under **d** 808 nm and **e** 1064 nm laser exposure for 10 min ($n = 5$ biologically independent samples). Data are presented as mean values ± SD. n.s.: not significant. $p > 0.05$; *$p < 0.05$; **$p < 0.01$; ***$p < 0.001$, analyzed by Student's two-sided test. **f**, **g** Fluorescence images of 4T1 cells after treated with various disposes. The cells were stained by **f** calcein-AM/PI (scale bars: 100 μm) and **g** DAPI/iFluor 488 (scale bars: 20 μm) before the analysis by CLSM. **h₁₋₈** Apoptosis profiles of 4T1 cells after treated with various disposes (I: PBS; II: NIR-I laser; III: NIR-II laser; IV: Nano-BFF; V: Nano-BFF + NIR-I laser; VI: Nano-BFF + NIR-II laser).

The PTT effect of Nano-BFF was then validated by characterizing the morphological change of actin filaments. Well-organized stress fibers with polymerized and stable structures were clearly visualized in the treatment groups of PBS, Nano-BFF, NIR-I laser or NIR-II laser. For the groups treated with Nano-BFF + NIR-I laser or Nano-BFF + NIR-II laser, the actin filament skeleton deformed severely, suggesting that the heating effect of Nano-BFF could significantly damage the actin filament polymerization (Fig. 3g). The apoptosis assay of 4T1 cells from various treatment groups was performed to quantitatively investigate the cell apoptosis induced by photonic hyperthermia. Laser irradiation or Nano-BFF administration under the defined conditions led to negligible apoptosis, whereas the groups treated with Nano-BFF followed by NIR-I and NIR-II laser irradiation caused approximately 91.4% and 94.5% apoptotic cells, respectively, indicating that NIR-induced photothermal effect of Nano-BFF resulted in the cell death dominantly via apoptosis rather than necrosis (Fig. 3h and Supplementary Fig. 16).

**In vivo biosafety and PA imaging of Nano-BFF.** The in vivo toxicity of Nano-BFF was evaluated to investigate its biosafety for potential clinical applications. Significantly, there was no marked difference between PBS-treated group and Nano-BFF-treated group in all of the blood indexes, demonstrating that Nano-BFF had no negative impact on the blood biochemistry

(Supplementary Fig. 17a–p and Supplementary Note 1). Furthermore, hematoxylin & eosin (H&E) staining of the main organs of the mice exhibited no obvious signal of tissue damage or inflammation lesion in all treatment groups, further confirming the excellent biocompatibility of as-designed Nano-BFF (Supplementary Fig. 17q). In addition, the in vivo blood circulation of Nano-BFF was investigated by monitoring the concentration of Nano-BFF in blood at various time points after intravenous administration. The concentration of Nano-BFF decayed rapidly at the initial stage and the half-life of Nano-BFF was calculated to be 1.41 h, indicating its excellent in vivo pharmacokinetic performance (Fig. 4a). Furthermore, in vivo biodistribution analysis was conducted to evaluate the accumulation of Nano-BFF in the main organs and tumor tissues at different time points post injection. Supplementary Fig. 18 illustrates that Nano-BFF mainly distributed in the liver tissues of mice. The accumulation of Nano-BFF at tumor sites was relatively high with the amounts of 4.46% ID g$^{-1}$ at 4 h and 8.57% ID g$^{-1}$ at 12 h, respectively, which should be ascribed to the typical enhanced permeability and retention effect and prolonged blood circulation time of Nano-BFF.

Owing to prominent photonic hyperthermia performance of Nano-BFF in both NIR-I and NIR-II biowindow, in vitro PA imaging of Nano-BFF at varied concentrations was recorded (Fig. 4b, c). A good linearity was observed between the intensity of PA signal and the dose of Nano-BFF at 900 and 1200 nm,

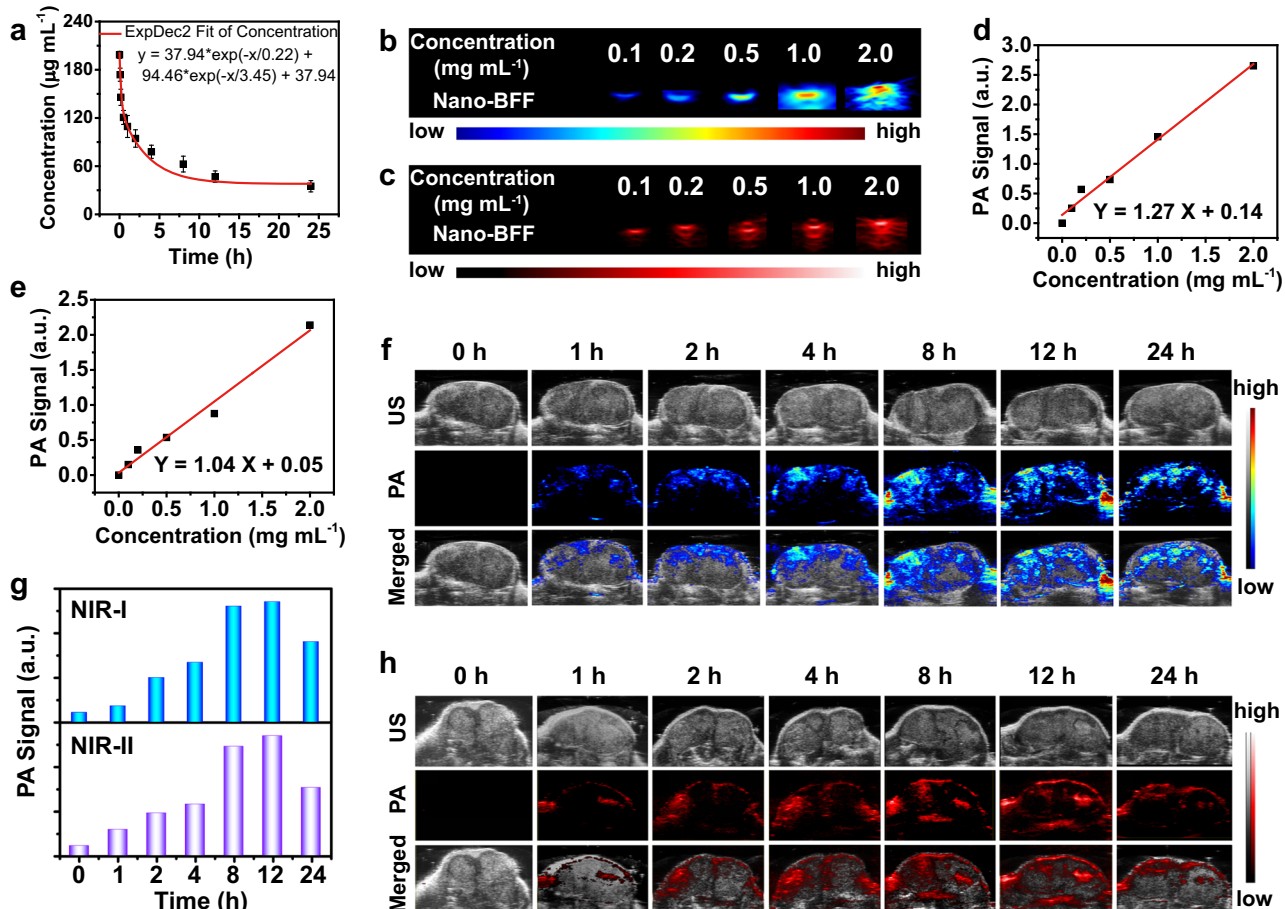

**Fig. 4 In vivo pharmacokinetic analysis and PA imaging of Nano-BFF. a** Blood circulation profile of Nano-BFF after intravenous administration ($n = 3$ biologically independent samples). Data are presented as mean values ± SD. **b, c** PA images of Nano-BFF under excitation at **b** 900 nm and **c** 1200 nm at varied concentrations. **d, e** Plot of PA amplitudes using Nano-BFF at **d** 900 nm and **e** 1200 nm as a function of the dose of Nano-BFF. **f–h** In vivo PA images of tumors in the **f** NIR-I and **h** NIR-II biowindow, as well as **g** corresponding PA values after intravenous injection of Nano-BFF (20 mg kg$^{-1}$) at various time points.

indicating desirable PA imaging capability of Nano-BFF in both NIR-I and NIR-II biowindow (Fig. 4d, e). Furthermore, the in vivo PA imaging performance of various doses of Nano-BFF was investigated by recording PA signals at tumor sites after varied time intervals post injection. The gradually increased PA signals of Nano-BFF at tumor regions in both NIR-I and NIR-II biowindow were observed by prolonging the incubation time, and the maximum signal intensity was detected at 12 h post injection on account of time-dependent accumulation of Nano-BFF at tumor sites within 12 h and its subsequent influence on the tumor microenvironment (Fig. 4f–h, Supplementary Fig. 19, and Supplementary Note 2). The desirable in vivo PA imaging capability indicates the feasibility of Nano-BFF as a contrast agent for cancer treatment under the guidance of PA imaging.

**In vivo photonic tumor hyperthermia in the NIR-II biowindow based on Nano-BFF.** Encouraged by the excellent in vitro photothermal conversion capability of Nano-BFF, its in vivo photonic hyperthermia efficacy was investigated (Fig. 5a). The mice bearing

4T1 tumors were randomly divided into six groups, including group I: PBS (control), group II: NIR-I laser, group III: NIR-II laser, group IV: Nano-BFF, group V: Nano-BFF + NIR-I laser, and group VI: Nano-BFF + NIR-II laser. Tumors were irradiated by NIR-I or NIR-II laser for 10 min at 12 h post injection. An IR thermal camera was employed to monitor in situ thermal imaging and real-time temperature of tumor regions upon the laser exposure. For mice administrated with Nano-BFF, the temperature at the tumor sites quickly increased by 18.4 °C and 19.8 °C during the first 6 min and then maintained at 55.8 °C and 57.5 °C within 10 min after NIR-I and NIR-II lasers irradiation, respectively (Fig. 5b, c). In comparison, upon the irradiation with NIR-I and NIR-II lasers, the temperature of the tumors in the PBS group showed a slight increase, and the final temperature reached 43.2 °C and 44.1 °C, respectively, demonstrating that Nano-BFF is capable of significantly elevating the local tumor temperature upon the NIR irradiation in both NIR-I and NIR-II biowindow. Relative tumor volumes of mice received various treatments were monitored to evaluate the in vivo photothermal ablation efficacy (Fig. 5d and Supplementary Fig. 20). Nano-BFF-mediated

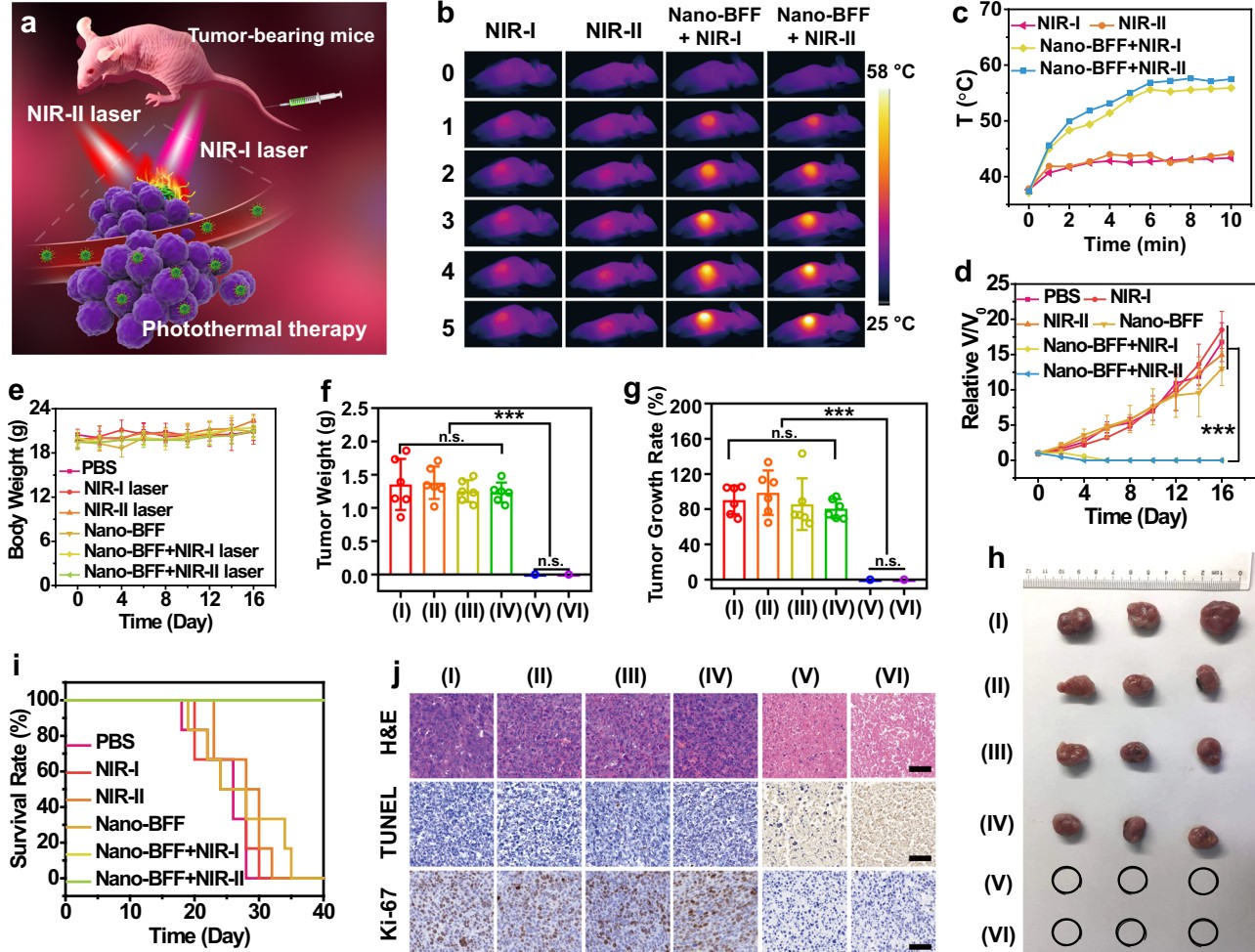

**Fig. 5 In vivo photonic tumor hyperthermia based on Nano-BFF in both NIR-I and NIR-II biowindow. a** Scheme of photonic cancer hyperthermia induced by Nano-BFF. **b** IR thermal images, and **c** tumor temperature variations of the mice in different treatment groups (NIR-I laser, NIR-II laser, Nano-BFF + NIR-I laser, and Nano-BFF + NIR-II laser) (n = 6 biologically independent samples). **d** Relative tumor volumes, **e** body weight variations of the mice, **f** tumor weights, and **g** tumor growth rates of 4T1 tumor-bearing mice after various treatments (n = 6 biologically independent samples). **h** Representative photographs of dissected tumors and **i** survival rates of the mice bearing 4T1 tumors after different treatment. Data are presented as mean values ± SD. n.s.: not significant. p > 0.05; *p < 0.05; **p < 0.01; ***p < 0.001, analyzed by Student's two-sided test. **j** H&E, TUNEL, and Ki-67 immunofluorescence staining of the tumor tissues after various treatments (I: PBS; II: NIR-I laser; III: NIR-II laser; IV: Nano-BFF; V: Nano-BFF + NIR-I laser; VI: Nano-BFF + NIR-II laser). Scale bars: 50 μm. A representative image of three biological replicates from each group is shown.

photothermal treatment in both NIR-I and NIR-II biowindow achieved complete tumor elimination with the survival rate of 100%, and no tumor regrowth occurred during the observation period of 40 days, validating that Nano-BFF could remarkably suppress tumor growth under the laser irradiation (Fig. 5f–i). In a sharp contrast, PBS injection, NIR laser irradiation only, or Nano-BFF administration without the laser irradiation exhibited no significant effect on the tumor suppression (Fig. 5f–h). In addition, the body weights of tumor-bearing mice were not influenced by these treatments, suggesting that photothermal treatment induced by Nano-BFF features high therapeutic biosafety (Fig. 5e). Representative H&E images of main organs exhibited negligible pathological damage or inflammatory signals during the whole treatment period, also demonstrating negligible adverse effect of Nano-BFF on normal tissues (Supplementary Fig. 21).

To verify the antitumor mechanism of Nano-BFF upon laser irradiation (808 nm or 1064 nm), H&E and terminal deoxynucleotidyl transferase uridine triphosphate nick end labeling (TUNEL) staining were conducted at 24 h after various treatments. Severe tumor cell necrosis/apoptosis was detected for groups of Nano-BFF + NIR-I laser and Nano-BFF + NIR-II laser, whereas other four groups (PBS, NIR-I laser, NIR-II laser, and Nano-BFF) revealed negligible tumor cell damage (Fig. 5j). Furthermore, the in vivo proliferative activity of varied treatment groups was assessed by Ki-67 antibody staining assay. In marked comparison with the treated groups of PBS, NIR-I laser, NIR-II laser, and Nano-BFF, the integration of Nano-BFF with NIR-I laser or NIR-II laser displayed substantial inhibition on the tumor cell proliferative activity (Fig. 5j).

In addition, immune competent 4T1 tumor-bearing BALB/c mouse model was also established to evaluate the photonic hyperthermia effect of Nano-BFF in vivo. The results further validated the prominent PTT efficacy of Nano-BFF in vivo (Supplementary Figs. 22 and 23 and Supplementary Note 3). Furthermore, the in vivo antitumor efficacy of Nano-BFF with various doses was also evaluated. As showed in Supplementary Fig. 24 (Supplementary Note 4), after exposed to NIR-II laser irradiation, the tumor temperature increased by 14.1 and 17.2 °C at the injection doses of 5 and 10 mg kg$^{-1}$ respectively, indicating excellent hyperthermia effect from Nano-BFF upon NIR-II laser irradiation. Importantly, Nano-BFF (5 mg kg$^{-1}$) plus NIR-II laser exposure caused obvious tumor suppression with the tumor growth rate of 67.6%. Complete tumor ablation without further recurrence was observed for the mice receiving Nano-BFF (10 mg kg$^{-1}$) administration and NIR-II laser irradiation. These findings confirmed that Nano-BFF could serve as an efficient photothermal agent with excellent biocompatibility.

**In vivo deep-tissue photonic tumor hyperthermia in the NIR-II biowindow based on Nano-BFF**. To confirm the superiority of the NIR-II biowindow over the NIR-I biowindow for in vivo deep-tissue photonic tumor hyperthermia, Nano-BFF was intravenously administered into nude mice bearing 4T1 tumors, and subsequently laser exposure was performed by penetrating through 4 mm of chicken breast tissue for mimicking the deep-tissue surroundings (Fig. 6a). After exposed to NIR-II laser illumination, the tumor temperature for the Nano-BFF-administrated mice reached to 52 °C (Fig. 6b, c), which was distinctly higher than that of Nano-BFF-administrated mice irradiated with NIR-I laser (45 °C) and PBS-treated mice exposed to NIR-II laser irradiation (42 °C). Correspondingly, the tumor growth of Nano-BFF-administrated mice was successfully suppressed throughout the observation period, and the tumors were completely ablated at 6th day after NIR-II laser irradiation. In a

remarkable contrast, the tumors of the mice treated with Nano-BFF and NIR-I laser under the same conditions grew rapidly, similar to that of the mice administrated with PBS (Fig. 6d, e). Furthermore, much higher deep-tissue therapeutic efficiency of Nano-BFF in the NIR-II biowindow over the NIR-I biowindow was verified by the H&E, TUNNEL, and Ki-67 staining (Fig. 6g). In addition, no substantial weight variations of the mice and no damage signal of main organs of the mice were observed for all treatment groups, demonstrating high biosafety of Nano-BFF administration and laser irradiation (Fig. 6f, h).

Based on the excellent accumulation of Nano-BFF in the liver tissues of mice and deep-penetration capability of NIR-II laser, an orthotopic HCC model was established to confirm the in vivo deep-tissue therapeutic efficacy of Nano-BFF under NIR-II laser irradiation. Tumor-bearing mice with uniform bioluminescence signals were randomly allocated into four groups, including PBS (control), Nano-BFF, Nano-BFF + NIR-I laser, and Nano-BFF + NIR-II laser. Tumor volumes of the mice were examined using bioluminescence imaging in vivo (Fig. 7a). In comparison with the PBS, Nano-BFF, and Nano-BFF + NIR-I laser groups, the Nano-BFF + NIR-II laser group exhibited a significantly higher suppressive effect on orthotopic liver tumor proliferation of the mice (Fig. 7b–f). The tumors in the Nano-BFF + NIR-II laser group had the minimum volumes among all treatment groups, which were in accordance with the results of the bioluminescence imaging in vivo (Fig. 7h). The antitumor performance was further evaluated via H&E, TUNEL, and Ki-67 staining of the representative tumor tissues. The largest region of apoptosis/necrosis and the least proliferation were detected in the tumor tissues of the mice receiving Nano-BFF administration and subsequent NIR-II laser irradiation, but no evident damage and apoptosis was observed in the normal liver tissues (Fig. 7i). Meanwhile, negligible body weight loss could be observed in each treatment group, which validated that Nano-BFF had no obvious toxicity toward the mouse health (Fig. 7g). In addition, H&E staining of the major organs exhibited no sign of toxicity in vivo (Supplementary Fig. 25). These results demonstrated that Nano-BFF achieved effective hyperthermia effect under NIR-II laser irradiation for deep-seated tumor ablation.

## Discussion
In this work, we have demonstrated an ultrastable NIR-II-absorbing organic nanomedicine for PA imaging-guided deep-tissue photo-to-heat energy-converting theranostics in the NIR-II biowindow. In comparison with the absorption of the BFF monomer in the NIR-I region, the self-assembled Nano-BFF reveals red-shifted absorption into the NIR-II region on account of strong π–π stacking interaction, allowing for higher MPE limit and deeper penetration depth in photonic hyperthermia under the guidance of PA imaging in the NIR-II biowindow. Noteworthy, Nano-BFF exhibits superior stability with respect to photothermal stability, photobleaching resistance, and resistance to aggressive agents, which is much better than that of FDA-approved ICG. Both in vitro and in vivo studies manifested that PA imaging-guided deep-tissue photonic hyperthermia in the NIR-II window could markedly ablate tumors without any recurrence. Consequently, Nano-BFF holds a great promise as a distinct theranostic agent for treating deep-seated tumors, and the present study provides a promising strategy to explore organic NIR-II-absorbing nanostructures for phototheranostic applications.

## Methods
**Preparation of Nano-BFF**. In brief, BFF (1 mg mL$^{-1}$) and F127 surfactant (30 mg) were dissolved in CH$_2$Cl$_2$ (1 mL) in a glass vial (20 mL). Deionized water was subsequently added to the glass vial and the mixture was sonicated via an ultrasonic probe for 1 h. Aqueous solution containing Nano-BFF nanoparticles was

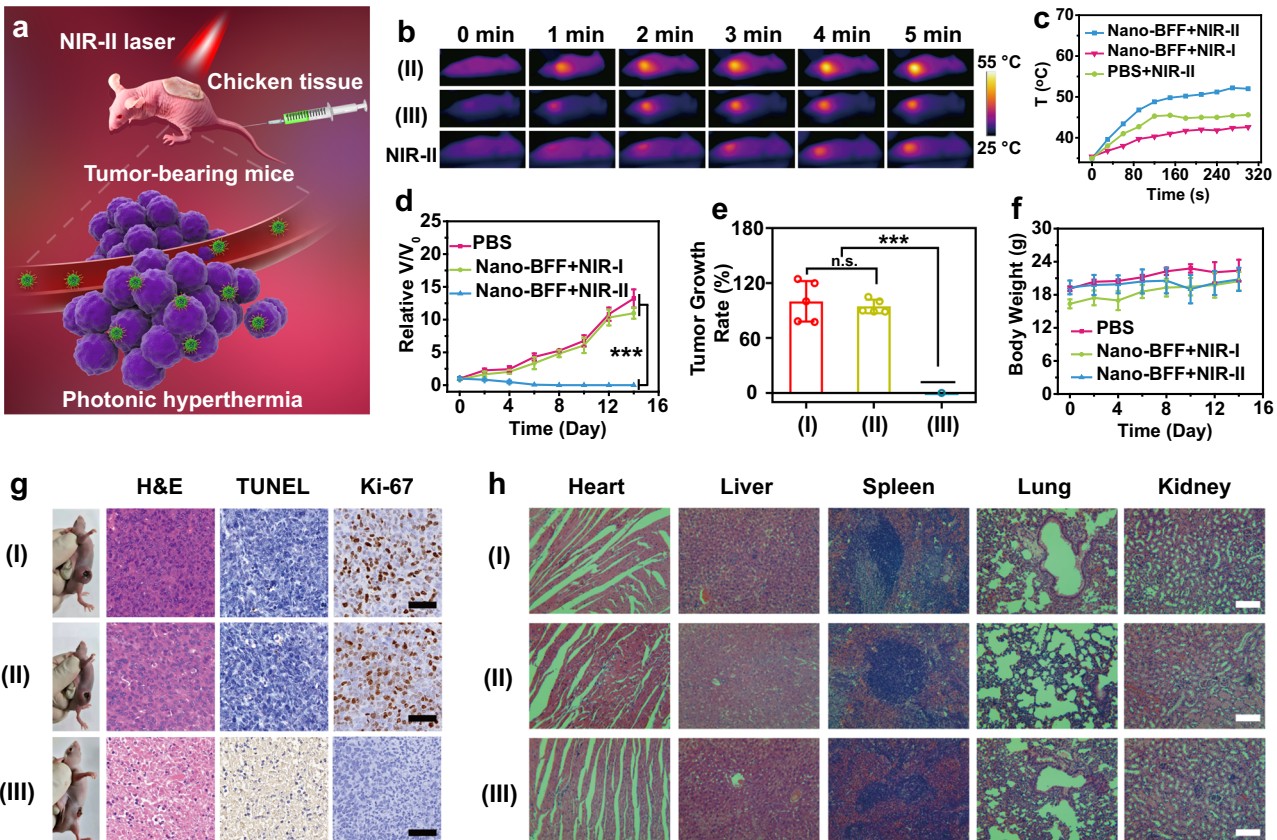

**Fig. 6 In vivo deep-tissue photonic tumor hyperthermia of Nano-BFF in the NIR-II biowindow. a** Schematic interpretation of deep-tissue photonic cancer hyperthermia using Nano-BFF. **b** Thermal images, and **c** temperature profiles at tumor regions of tumor-bearing mice in different treatment groups after the penetration through 4 mm of chicken breast tissue ($n = 5$ biologically independent samples). **d** Relative tumor volumes, **e** tumor growth rates, and **f** body weight variations of the mice in varied treatment groups ($n = 5$ biologically independent samples). Data are presented as mean values ± SD. n.s.: not significant. $p > 0.05$; $*p < 0.05$; $**p < 0.01$; $***p < 0.001$, analyzed by Student's two-sided test. **g** Representative photographs of mice and corresponding H&E, TUNEL, and Ki-67 antibody staining of the tumor tissues after various treatments (scale bars: 50 µm). **h** H&E staining of main organs from varied treatment groups (I: PBS; II: Nano-BFF + NIR-I laser; III: Nano-BFF + NIR-II laser, penetration depth: 4 mm). Scale bars: 100 µm. A representative image of three biological replicates from each group is shown.

obtained after complete evaporation of organic solvents. As-prepared Nano-BFF nanoparticles were dialyzed for 48 h to get rid of any impurities.

**Evaluation of photothermal conversion performance**. The photothermal performance of Nano-BFF was assessed by exposing the Nano-BFF solution to 808 nm and 1064 nm laser illumination at varied power densities. The real-time temperature and in situ thermal images of Nano-BFF aqueous solution at different concentrations were monitored using a thermal imaging system. The photostability of Nano-BFF was evaluated by monitoring the real-time temperature of Nano-BFF solution upon 808 nm and 1064 nm laser exposure over several cycles of heating/cooling processes. The photo-to-heat conversion efficiency of Nano-BFF was calculated[17].

**Cellular uptake studies**. 4T1 cells were incubated with Cy3-labeled Nano-BFF (50 µg mL$^{-1}$) for 1, 2, 4, and 12 h, respectively. After washed by PBS for twice, 4T1 cells were stained with 4′,6-diamidino-2-phenylindole (DAPI). Subsequently, the treated cells were imaged by CLSM (excitation wavelengths: 405 nm for DAPI and 550 nm for Cy3, respectively).

**Cellular cytotoxicity evaluation**. 4T1 cells were treated with various concentrations of Nano-BFF for 12 h and subsequently exposed to laser irradiation (808 nm or 1064 nm, 1 W cm$^{-2}$) for 10 min. Afterwards, the 4T1 cells were cultured for another 4 h. Cell survival rate of Nano-BFF was detected using a standard CCK-8 assay.

To visualize the live/dead cells, 4T1 cells were treated with Nano-BFF (100 µg mL$^{-1}$) for 12 h and subsequently exposed to laser irradiation (808 nm or 1064 nm, 1 W cm$^{-2}$) for 10 min. After incubation for 4 h, the cells were stained with calcein-AM/PI for 30 min. Finally, the labeled cells were rinsed twice with PBS and imaged by CLSM (excitation wavelengths: 488 nm and 530 nm for calcein-AM and PI, respectively).

**Cytoskeleton morphology studies**. 4T1 cells receiving various treatments were employed to visualize cellular cytoskeleton morphology. These treatments include (I) PBS (control), (II) 808 nm laser, (III) 1064 nm laser, (IV) Nano-BFF (100 µg mL$^{-1}$), (V) Nano-BFF (100 µg mL$^{-1}$) + 808 nm laser, and (VI) Nano-BFF (100 µg mL$^{-1}$) + 1064 nm laser. Laser irradiation (1 W cm$^{-2}$, 10 min) was applied in groups II, III, V, and VI. After these treatments, the cells were incubated for 4 h and rinsed twice with PBS. Afterwards, the cells were labeled with DAPI and Phalloidin-Alexa Fluor 488 for 30 min. The cytoskeleton morphology in various treatment groups was observed by CLSM (excitation wavelengths: 405 nm and 488 nm for DAPI and Phalloidin-Alexa Fluor 488, respectively).

**Cell apoptosis evaluation**. 4T1 cells were treated under different conditions to evaluate the cell apoptosis. The various treatments include (I) PBS (control), (II) 808 nm laser, (III) 1064 nm laser, (IV) Nano-BFF (100 µg mL$^{-1}$), (V) Nano-BFF (100 µg mL$^{-1}$) + 808 nm laser, and (VI) Nano-BFF (100 µg mL$^{-1}$) + NIR-II laser. Laser irradiation (1 W cm$^{-2}$, 10 min) was applied in groups II, III, V, and VI. After these treatments, the cells were incubated for 4 h and rinsed twice with PBS. The treated cells were then incubated for 4 h and subsequently harvested by trypsinization and centrifugation. The precipitated cells were dispersed with PBS and then labeled with Annexin V-FITC and PI for 15 min before flow cytometry analysis.

**In vivo toxicity evaluation**. Animal experiment methods were in accordance with the regulations of the Regional Ethics Committee for Animal Experiments approved by the Administrative Committee of Laboratory Animals of Shanghai Tenth People's Hospital, Tongji University School of Medicine. Kunming mice were randomized into 4 groups and the mice were subjected to intravenous injection of Nano-BFF (40 mg kg$^{-1}$). Afterwards, the mice were sacrificed at the 7th, 15th, and 30th day post injection. The blood and main organs of the mice were harvested for hematological analysis and histological examination.

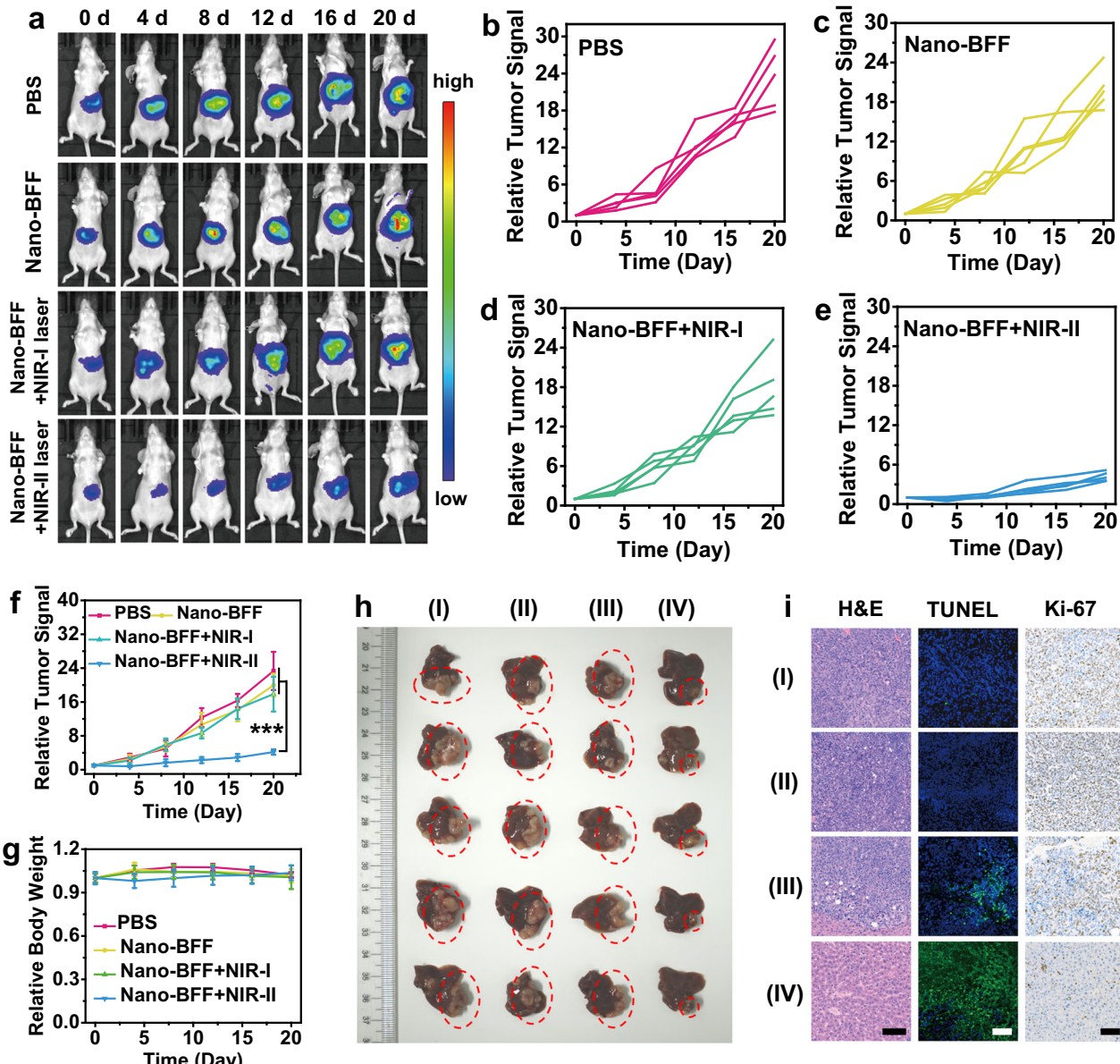

**Fig. 7 In vivo antitumor performance of Nano-BFF toward orthotopic HCC. a** Bioluminescence images of the mice treated with PBS, Nano-BFF, Nano-BFF + NIR-I laser, and Nano-BFF + NIR-II laser on days 0, 4, 8, 12, 16, and 20, respectively ($n = 5$ biologically independent samples). **b–e** Relative orthotopic liver tumor luminescence levels of mice after various treatments, including **b** PBS, **c** Nano-BFF, **d** Nano-BFF + NIR-I laser, and **e** Nano-BFF + NIR-II laser. **f** Relative orthotopic liver tumor luminescence levels of the mice, and **g** body weight variations of the mice in various treatment groups within 20 days ($n = 5$ biologically independent samples). Scale bars: 100 μm. Data are presented as mean values ± SD. $p > 0.05$; *$p < 0.05$; **$p < 0.01$; ***$p < 0.001$, analyzed by Student's two-sided test. **h** Photographs of liver tissues in each treatment group, where the orthotopic tumor tissues were highlighted with red dotted circles. **i** Histological images of the orthotopic liver tumor tissues using H&E, TUNEL, and Ki-67 staining assays (I: PBS; II: Nano-BFF; III: Nano-BFF + NIR-I laser; IV: Nano-BFF + NIR-II laser). A representative image of three biological replicates from each group is shown.

**Pharmacokinetics evaluation.** The pharmacokinetics of Nano-BFF were evaluated in healthy Kunming mice ($n = 3$). The blood of mice was obtained at varied duration after intravenous administration of Nano-BFF ($20\,mg\,kg^{-1}$). The doses of Nano-BFF in the blood at various time points were determined to evaluate the pharmacokinetic behavior.

**In vivo biodistribution of Nano-BFF.** The mice bearing 4T1 tumors were intravenously administrated with Nano-BFF ($20\,mg\,kg^{-1}$). Subsequently, the mice were killed at designated time points post injection, and the main organs and tumors were dissected and treated with chloroazotic acid. The distribution of Nano-BFF in each tissue was determined.

**PA imaging assessments.** The in vitro PA signals of Nano-BFF solution (0, 0.1, 0.2, 0.5, 1.0, and $2.0\,mg\,mL^{-1}$) at various concentrations were collected with the excitation wavelengths at 900 nm and 1200 nm. To investigate in vivo PA imaging

after intravenous administration with Nano-BFF (5, 10, and $20\,mg\,kg^{-1}$), the in vivo PA images and signal intensities at tumor sites of the mice were monitored at various time intervals post injection.

**In vivo photothermal cancer treatment.** The nude mice or immune competent BALB/c mice bearing 4T1 tumors were allocated into 6 groups, including (I) PBS, (II) 808 nm laser, (III) 1064 nm laser, (IV) Nano-BFF ($20\,mg\,kg^{-1}$), (V) Nano-BFF ($20\,mg\,kg^{-1}$) + 808 nm laser, and (VI) Nano-BFF ($20\,mg\,kg^{-1}$) + 1064 nm laser. Laser irradiation ($1\,W\,cm^{-2}$, 10 min) was applied in groups II, III, V, and VI. In addition, the in vivo PTT efficacy of Nano-BFF at the doses of 5 and $10\,mg\,kg^{-1}$ under NIR-II laser irradiation ($1\,W\,cm^{-2}$, 10 min) was also evaluated. The body weights and tumor volumes were monitored every other day after various treatments. The tumors of represented mice were collected at 24 h after different disposes for histological analysis to investigate the therapeutic effect in different treatment groups. At the end of the whole experiment, the main organs from

represented mice in various groups were harvested for H&E staining to validate the biosafety of Nano-BFF.

**In vivo deep-tissue photothermal cancer treatment**. The mice bearing 4T1 tumors were randomly separated into three groups. The treatment groups include (I) PBS, (II) Nano-BFF (20 mg kg$^{-1}$) + 808 nm laser, and (III) Nano-BFF (20 mg kg$^{-1}$) + 1064 nm laser. Laser irradiation (1 W cm$^{-2}$) in groups II and III was performed through 4 mm chicken breast tissue for 10 min. After various treatments, the body weights of the mice and tumor volumes were monitored every other day. The represented mice were killed at 24 h after different disposes, and corresponding tumors were harvested for histological analysis to assess the therapeutic effect. In addition, the main organs were collected at the end of the whole experiment for H&E staining.

**In vivo photothermal treatment of orthotopic HCC**. To assess orthotopic HCC suppressive efficacy of Nano-BFF after NIR-II laser exposure, the establishment of orthotopic HCC model was confirmed by bioluminescence imaging. Tumor-bearing mice with uniform bioluminescence signal were randomized into following four groups, including PBS, Nano-BFF, Nano-BFF + 808 nm laser, and Nano-BFF + 1064 nm laser. The administration dose of Nano-BFF was 20 mg kg$^{-1}$. Laser irradiation (1 W cm$^{-2}$, 10 min) was applied. The bioluminescence images and intensity of orthotopic HCC bearing mice were detected every 4 days to monitor the tumor growth in each treatment group. Meanwhile, body weights of the mice in each treatment group were also monitored. The liver of represented mice in each treatment group was harvested at 24 h for histological analysis to assess the treatment efficacy of Nano-BFF upon NIR-II laser exposure. At the end of the experiment, the livers and other major organs were harvested for H&E staining.

**Statistical analysis**. All data were presented as mean ± standard deviation unless otherwise stated. The statistical significance was determined using a two-sided Student's test (n.s.: not significant, *$P < 0.05$, **$P < 0.01$, and ***$P < 0.001$).

**Reporting summary**. Further information on research design is available in the Nature Research Reporting Summary linked to this article.

## Data availability
All the other data supporting the findings of this study are available within the article and its supplementary information files and from the corresponding author upon reasonable request. A reporting summary for this article is available as a Supplementary Information file.

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

## Acknowledgements

We greatly acknowledge the financial support from the National Key R&D Program of China (grant no. 2016YFA0203700), National Natural Science Foundation of China (grant no. 51722211, 51672303, and 51902336), Program of Shanghai Subject Chief Scientist (grant no. 18XD1404300), Development Fund for Shanghai Talents (grant no. 2018114) and China Postdoctoral Science Foundation (grant no. 2018M642097). This research is also supported by the Singapore National Research Foundation Investigatorship (grant no. NRF-NRFI2018-03) and the Singapore Academic Research Fund (grant no. TR12/19).

## Author contributions

Y.Z., Y.C., and H.X. conceived and designed the experiments. H.X., L.Z., L.Y., H.C., and C.W. performed the experiments and analyzed the results. Y.Z., Y.C., and H.X. wrote and revised the manuscript.

## Competing interests

The authors declare no competing interests.
