## [Peer Review File · Nature Communications]

Reviewers' comments:

Reviewer #1 (Remarks to the Author):

Xiang et al developed an effective NIR-II nano-theranostic agent via the assembling of the NIR-I absorbing agent with amphiphilic biocompatible polymer for combating tumor progression with high deep tissue penetration capability. This study is well designed and there are relatively solid evidences to support the hypothesis of authors. However, several critical issues still need to be addressed to enhance the quality of it.

1. Authors systematically evaluated the stabilities of the developed nano-agent involving photothermal stability, photobleaching resistance and physiological stability under oxidative condition. As is known, acidic microenvironment as one of the typical hallmarks of tumor plays critical roles for the tumor progression and interferes with the biological behavior of drugs within tumor. Therefore, I suggest authors to evaluate the stability under acidic conditions.
2. In this study, 4T1 tumor-bearing nude mice were utilized for evaluating the therapeutic efficacy of the nano-theranostic agent. Actually, there are extensive evidences have demonstrated photothermal therapy can also induce antitumor immune responses besides directly damaging tumors. Meanwhile, 4T1 mouse breast tumor cells are commonly used for syngeneic tumor model. In other words, the immune competent mouse 4T1 tumor may be a better model for evaluating the treatment efficacy of the nano-agent. Could authors explain why select nude mice for this work?
3. To confirm the unique advantage of NIR-II nanotheranostic agent, authors adopted chicken breast tissue on the tumor for mimicking the deep tissue condition. It is no doubt that the real deep-tissue surroundings are big different from that of chicken breast tissue covering on the subcutaneous tumor. Therefore, I strongly recommend authors to provide related data using orthotopic tumor model like liver cancer or pancreatic cancer. It will be greatly useful to support the hypothesis of this work.

Reviewer #2 (Remarks to the Author):

A self-assembled organic nanoparticle (Nano-BFF) was reported to possess a broad absorption spectrum spanning over both NIR I and NIR II regions. Nano-BFF shows high photo stability and efficient photothermal conversion. The authors evaluated its use as a theranostic material for photoacoustic imaging and hyperthermia cancer therapy. While the data demonstrate promising properties of this new nanomaterial, its advantages over similar materials have not been conclusively shown.

First, the use of ICG for comparison may not be sufficient. There are reported nanoparticles including polymer nanoparticles, at least they should compare to gold nano rods widely used for photoacoustic imaging and photothermal therapy. They are readily available. I could not find the extinction coefficient of Nano-BFF, in trying to get a sense of how much better this material was.

The authors did a proof-of-principle experiment to show the depth penetration effect by using chicken breast tissue. It would be more preferred a second deep-tissue tumor model be used for the demonstration of the photothermal effect.

Figure 2I shows that at 4 mm penetration depth, the ΔT value for 808 nm is approximately 4 degree less than that for 1064 nm. In Figure 6c however 1064nm seems to produce 7 degree higher than 808 nm even in vivo (Page 9 line 270-271). These results have to be reconciled.

Figure 2a&c vs. e&g appear to indicate that the photothermal efficiency at 808 nm is higher than at 1064 nm: the concentration was the same in both cases, but the temperature was apparently higher at 808 nm than 1064 nm. More details are required to show the calculation for 28.6% for 808 nm and 343.3% for 1064 nm.

No biodistribution data of the nanoparticles after iv injection. How many tumor uptakes?

The injection dose was 20 mg/kg for both photo acoustic imaging and photo thermal therapy. This is quite high. Did the authors try lower concentrations?

The proton NMR spectra of formazn and BFF appear to indicate the presence of some impurity in the sample.

Figure 6C; the green line should be Nano-BFF+NIR-I stead of PBS+NIR-II.

No statistical analysis in the work. MPE should be calculated in the phototherapy experiments.

Response to Reviewer 1#'s Comments:

Xiang et al developed an effective NIR-II nano-theranostic agent via the assembling of the NIR-I absorbing agent with amphiphilic biocompatible polymer for combating tumor progression with high deep tissue penetration capability. This study is well designed and there are relatively solid evidences to support the hypothesis of authors. However, several critical issues still need to be addressed to enhance the quality of it.

Response: Thanks for the useful comments. We have revised the manuscript according to your constructive comments and suggestions. Please refer to the following point-by-point response.

1. Authors systematically evaluated the stabilities of the developed nano-agent involving photothermal stability, photobleaching resistance and physiological stability under oxidative condition. As is known, acidic microenvironment as one of the typical hallmarks of tumor plays critical roles for the tumor progression and interferes with the biological behavior of drugs within tumor. Therefore, I suggest authors to evaluate the stability under acidic conditions.

Response: Thank you for this constructive suggestion. Based on your suggestion, we have performed dynamic light scattering (DLS) analysis to assess the stability of Nano-BFF under acidic physiological condition. The particle size of Nano-BFF at pH 5.5 and pH 6.5 remained stable without obvious fluctuation over 24 h, indicating high stability of Nano-BFF under acidic condition. We have clarified this issue and added the related data in the revised manuscript (**Page 4**) and supplementary information (**Supplementary Fig. 12**) according to your kind suggestion.

Supplementary Fig. S12. Hydrodynamic diameter of Nano-BFF in PBS (pH = 7.4, 5.5, and 6.5) solutions for 1, 4, 12, and 24 h.

2. In this study, 4T1 tumor-bearing nude mice were utilized for evaluating the therapeutic efficacy of the nano-theranostic agent. Actually, there are extensive evidences have demonstrated photothermal therapy can also induce antitumor immune responses besides directly damaging tumors. Meanwhile, 4T1 mouse breast tumor cells are commonly used for syngeneic tumor model. In other words, the immune competent mouse 4T1 tumor may be a better model for evaluating the treatment efficacy of the nano-agent. Could authors explain why select nude mice for this work?

Response: Thank you for this valuable comment. According to the results of photothermal therapy studies on 4T1 tumor-bearing nude mice in various treatment groups, no significant effect on the tumor inhibition was observed in the control treatment groups (PBS injection, NIR-I laser irradiation, NIR-II laser irradiation, and Nano-BFF administration). Nano-BFF-mediated photothermal treatment in both NIR-I and NIR-II biowindows achieved complete tumor elimination, validating that Nano-BFF could remarkably suppress tumor growth under the laser irradiation. In addition, 4T1 breast tumor model in nude mice has been widely used to investigate the photothermal efficacy of various photothermal agents (e.g., *Nat. Commun.* **11**, 1857 (2020); *J. Am. Chem. Soc.* **142**, 1735-1739 (2020); *Adv. Mater.* **32**, 2000542 (2020); *Adv. Mater.* **30**, 1705980 (2018)).

Based on your kind suggestion, immune competent 4T1 tumor-bearing BALB/c mouse model was also established to evaluate the photonic hyperthermia effect of Nano-BFF *in vivo*. The mice were randomly divided into 6 groups: PBS (control), NIR-I laser, NIR-II laser, Nano-BFF, Nano-BFF + NIR-I laser, and Nano-BFF + NIR-II laser. The tumors were exposed to laser irradiation (808 nm or 1064 nm, 1 W cm⁻²) for 10 min at 12 h post-injection. Real-time temperature and *in-situ* thermal images were monitored by an infrared imaging system to visualize the temperature elevation under NIR-I or NIR-II laser irradiation (**Supplementary Fig. 22a**). As presented in **Supplementary Fig. 22b**, the temperature of Nano-BFF + NIR-I laser and Nano-BFF + NIR-II laser groups elevated quickly by 19.9 °C and 18.1 °C during the first 6 min and then increased slowly to 55.1 °C and 54.0 °C within 10 min, respectively. In contrast, the groups of NIR-I laser and NIR-II laser exhibited slight increase in temperature, with the final temperature of 43.1 °C and 42.5 °C respectively, which demonstrated that Nano-BFF exhibited excellent capability in elevating tumor temperature upon NIR-I laser or NIR-II laser irradiation. Tumor volumes and body weights in all treatment groups were recorded every 2 day. The groups with the injection of PBS and Nano-BFF without laser exposure revealed rapid tumor growth during the treatment period. In comparison, mice administrated with Nano-BFF and subsequent NIR-I laser or NIR-II laser irradiation achieved complete tumor suppression, validating that Nano-BFF exhibited high therapeutic efficacy *in vivo* in both NIR-I and NIR-II biowindows (**Supplementary Fig. 23a,c,d**). In addition, no

significant body weight loss of mice could be observed in all treatment groups, implying high biosafety of Nano-BFF *in vivo* (Supplementary Fig. 23b).

To further validate the enhanced therapeutic efficacy of Nano-BFF, the representative tumors of the mice in all treatment groups were collected for histological analysis using H&E, TUNEL, and Ki-67 antibody staining. Large areas of necrosis/apoptosis were observed on the tumor slices for the groups of Nano-BFF + NIR-I laser and Nano-BFF + NIR-II laser, which were not obvious for that of control groups of PBS, NIR-I laser, NIR-II laser, and Nano-BFF (Supplementary Fig. 23e). These results further verified the prominent photothermal therapeutic efficacy of Nano-BFF *in vivo*. We have clarified this issue and added the related data in the revised manuscript (Page 10) and supplementary information (Supplementary Fig. 22 and 23).

Supplementary Fig. 22. (a) IR thermal images, and (b) temperature variations at the tumor sites of 4T1 tumor-bearing BALB/c mice in various treatment groups, including NIR-I laser, NIR-II laser, Nano-BFF + NIR-I laser, and Nano-BFF + NIR-II laser groups.

Supplementary Fig. 23. (a) Relative tumor volumes, (b) body weights of the mice, (c) tumor weights, (d) photographs of tumors, and (e) H&E, TUNEL, and Ki-67 staining of

tumor tissues in various treatment groups (I: PBS; II: NIR-I laser; III: NIR-II laser; IV: Nano-BFF; V: Nano-BFF + NIR-I laser; VI: Nano-BFF + NIR-II laser). Scale bars: 50 μ m.

3. To confirm the unique advantage of NIR-II nanotheranostic agent, authors adopted chicken breast tissue on the tumor for mimicking the deep tissue condition. It is no doubt that the real deep-tissue surroundings are big different from that of chicken breast tissue covering on the subcutaneous tumor. Therefore, I strongly recommend authors to provide related data using orthotopic tumor model like liver cancer or pancreatic cancer. It will be greatly useful to support the hypothesis of this work.

Response: Thank you for this valuable suggestion. Based on the efficient accumulation of Nano-BFF in the liver tissues of mice and deep-penetration capability of NIR-II laser, an orthotopic hepatocellular carcinoma (HCC) model was established to confirm the *in vivo* deep-tissue therapeutic efficacy of Nano-BFF under NIR-II laser irradiation. Tumor-bearing mice with uniform bioluminescence signals were randomly divided into 4 groups, PBS (control), Nano-BFF, Nano-BFF + NIR-I laser, and Nano-BFF + NIR-II laser. Tumor volumes of the mice were examined using bioluminescence imaging *in vivo* (**Fig. 7a**). In comparison with the PBS, Nano-BFF, and Nano-BFF + NIR-I laser groups, the Nano-BFF + NIR-II laser group exhibited a significantly higher suppressive effect on orthotopic liver tumor proliferation of the mice (**Fig. 7b-f**). The tumors in the Nano-BFF + NIR-II laser group had the minimum volumes among all treatment groups, which was in accordance with the results of the bioluminescence imaging *in vivo* (**Fig. 7h**). The anti-tumor performance was further evaluated *via* H&E, TUNEL, and Ki-67 staining of the representative tumor tissues. The largest region of apoptosis/necrosis and the least proliferation were detected in the tumor tissues of the mice receiving Nano-BFF administration and subsequent NIR-II laser irradiation, but no evident damage and apoptosis was observed in the normal liver tissues (**Fig. 7i**). Meanwhile, negligible body weight loss could be observed in each treatment group, which validated that Nano-BFF has no obvious toxicity toward the mice health (**Fig. 7g**). In addition, H&E staining of the major organs exhibited no sign of toxicity *in vivo* (**Supplementary Fig. 25**). These results demonstrated that Nano-BFF achieved effective hyperthermia effect under NIR-II laser irradiation for deep-seated tumor ablation. We have clarified this issue and added the related data in the revised manuscript (**Fig. 7; Page 11, and Page 15**) and supplementary information (**Supplementary Fig. 25**) according to your suggestion.

Fig. 7. *In vivo* antitumor performance of Nano-BFF toward orthotopic hepatocellular carcinoma. (a) Bioluminescence images of the mice treated with PBS, Nano-BFF, Nano-BFF + NIR-I laser, and Nano-BFF + NIR-II laser on days 0, 4, 8, 12, 16, and 20, respectively. (b-e) Orthotopic liver tumor luminescence levels of mice after various treatments, including (b) PBS, (c) Nano-BFF, (d) Nano-BFF + NIR-I laser, and (e) Nano-BFF + NIR-II laser. (f) Relative orthotopic liver tumor luminescence levels of the mice, and (g) Body weight variations of the mice in various treatment groups within 20 days. (h) Photographs of liver tissues in each treatment group, where the orthotopic tumor tissues were highlighted with red dotted circles ($n = 5$). (i) Histological images of the orthotopic liver tumor tissues in various treatment groups using H&E, TUNEL, and Ki-67 staining assays (I: PBS; II: Nano-BFF; III: Nano-BFF + NIR-I laser; IV: Nano-BFF + NIR-II laser). Scale bars: 100 μm .

Supplementary Fig. S25. H&E staining images of major organs from orthotopic liver tumor bearing mice in various treatment groups, including PBS, Nano-BFF, Nano-BFF + NIR-I laser, and Nano-BFF + NIR-II laser groups. Scale bars: 100 μ m.

Response to Reviewer 2#'s Comments:

A self-assembled organic nanoparticle (Nano-BFF) was reported to possess a broad absorptions spectrum spanning over both NIR I and NIR II regions. Nano-BFF shows high photo stability and efficient photothermal conversion. The authors evaluated its use as a theranostic material for photoacoustic imaging and hyperthermia cancer therapy. While the data demonstrate promising properties of this new nanomaterial, its advantages over similar materials have not been conclusively shown.

Response: Thank you for useful comments. We have performed additional experiments to address the issues raised by the reviewer. Please see the following detailed responses to your comments and suggestions.

1. First, the use of ICG for comparison may not be sufficient. There are reported nanoparticles including polymer nanoparticles, at least they should compare to gold nano rods widely used for photoacoustic imaging and photothermal therapy. They are readily available. I could not find the extinction coefficient of Nano-BFF, in trying to get a sense of how much better this material was.

Response: Thank you for your constructive question. To further validate the excellent photothermal stability of Nano-BFF, we evaluated the photostability of BFF, Nano-BFF, clinically used indocyanine green (ICG), and Au nanorods (Au NRs) under NIR-I or NIR-II laser irradiation. The characteristic peak of Au nanorods decreased and shifted to the shorter wavelength region, owing to the significant aggregation of Au nanorods upon exposure to continuous NIR-I laser irradiation (**Supplementary Fig. 10**). In contrast, the absorption spectra of BFF and Nano-BFF were nearly unchanged during NIR-I laser irradiation period (**Supplementary Fig. 11b-d,f**). Especially, NIR-II laser illumination exhibited a negligible effect on the optical property of Nano-BFF (**Supplementary Fig. 11e,f**). These results demonstrated that Nano-BFF with high photostability could serve as a desirable phototheranostic agent for PA imaging-guided disease treatment.

Furthermore, the photothermal conversion performance of Nano-BFF was also investigated. As presented in **Fig 2c,g**, the photothermal conversion efficiency of Nano-BFF was calculated to be 28.6% and 34.3% at 808 nm and 1064 nm respectively, which is significantly higher than that of various photothermal agents (ICG, 3.1%;⁴¹ IR1048-MZ, 20.2%;⁴³ Au NRs, 21%⁴⁴), demonstrating high potential photothermal conversion performance of Nano-BFF *in vivo*. We have clarified this issue and added the related data in the revised manuscript (**Fig. 2c,g; Page 4 and Page 5**) and supplementary information (**Supplementary Fig. 10**) according to your suggestion.

In addition, the absorbance of Nano-BFF at 808 nm and 1064 nm exhibited a gradual increase with elevated concentrations, and the molar extinction coefficient at 808 nm and 1064 nm was calculated to be 3.07×10^5 and $2.22 \times 10^5 \text{ M}^{-1} \text{ cm}^{-1}$, which was

comparable or higher than that of ICG ($2.16 \times 10^5 \text{ M}^{-1} \text{ cm}^{-1}$ at 795 nm)⁴¹ and Mito-CCy ($1.55 \times 10^5 \text{ M}^{-1} \text{ cm}^{-1}$ at 734 nm)⁴², indicating that Nano-BFF is a promising photoabsorbing agent to achieve PA imaging-guided photonic hyperthermia in the NIR-II biowindow (**Supplementary Fig. 9**). We have clarified this issue and added the related data in the revised manuscript (**Page 4**) and supplementary information (**Supplementary Fig. 9**).

References:

41. X. Zhao, S. Long, M. Li, J. Cao, Y. Li, L. Guo, W. Sun, J. Du, J. Fan, X. Peng, *J. Am. Chem. Soc.* **142**, 1510-1517 (2020).
42. H. S. Jung, J. H. Lee, K. Kim, S. Koo, P. Verwilt, J. L. Sessler, C. Kang, J. S. Kim, *J. Am. Chem. Soc.* **139**, 9972-9978 (2017).
43. X. Meng, J. Zhang, Z. Sun, L. Zhou, G. Deng, S. Li, W. Li, P. Gong, L. Cai, *Theranostics* **8**, 6025-6034 (2018).
44. J. Zeng, D. Goldfeld, Y. Xia, *Angew. Chem. Int. Ed.* **52**, 4169-4173 (2013).

Supplementary Fig. S10. (a) UV-vis-NIR absorption spectra of Au nanorods (NRs) and Au NRs after continuous NIR-I laser irradiation for 7 min. (b,c) TEM images of (b) Au NRs, and (c) Au NRs after continuous NIR-I laser irradiation for 7 min. Scale bars: 100 μm.

Supplementary Fig. 9. (a) UV-vis-NIR absorption spectra of various concentrations of Nano-BFF. (b) Molar extinction coefficient of Nano-BFF at 808 nm and 1064 nm.

2. The authors did a proof-of-principle experiment to show the depth penetration effect by using chicken breast tissue. It would be more preferred a second deep-tissue tumor model be used for the demonstration of the photothermal effect.

Response: Thank you for this valuable suggestion. Based on the efficient accumulation of Nano-BFF in the liver tissues of mice and deep-penetration capability of NIR-II laser, an orthotopic hepatocellular carcinoma model was established to confirm the *in vivo* deep-tissue therapeutic efficacy of Nano-BFF under NIR-II laser irradiation. Tumor-bearing mice with uniform bioluminescence signals were randomly divided into 4 groups: PBS (control), Nano-BFF, Nano-BFF + NIR-I laser, and Nano-BFF + NIR-II laser. Tumor volumes of the mice were examined using bioluminescence imaging *in vivo* (**Fig. 7a**). In comparison with the PBS, Nano-BFF, and Nano-BFF + NIR-I laser groups, the Nano-BFF + NIR-II laser group exhibited a significantly higher suppressive effect on orthotopic liver tumor proliferation of the mice (**Fig. 7b-f**). The tumors in the Nano-BFF + NIR-II laser group had the minimum volumes among all treatment groups, which was in accordance with the results of the bioluminescence imaging *in vivo* (**Fig. 7h**). The anti-tumor performance was further evaluated *via* H&E, TUNEL, and Ki-67 staining of the representative liver tumor tissues. The largest region of apoptosis/necrosis and the least proliferation were detected in the tumor tissues of the mice receiving Nano-BFF administration and subsequent NIR-II laser irradiation, but no evident damage and apoptosis was observed in the normal liver tissues (**Fig. 7i**). Meanwhile, negligible body weight loss could be observed in each treatment group, which validated that Nano-BFF has no obvious toxicity toward the mice health (**Fig. 7g**). In addition, H&E staining of the major organs exhibited no sign of toxicity *in vivo* (**Supplementary Fig. 25**). These results demonstrated that Nano-BFF achieved effective hyperthermia effect under NIR-II laser irradiation for deep-seated tumor ablation. We have clarified this issue and added the related data in the revised manuscript (**Fig. 7; Page 11, and Page 15**) and supplementary information (**Supplementary Fig. 25**) according to your suggestion.

Fig. 7. *In vivo* antitumor performance of Nano-BFF toward orthotopic hepatocellular carcinoma (HCC). (a) Bioluminescence images of the mice treated with PBS, Nano-BFF, Nano-BFF + NIR-I laser, and Nano-BFF + NIR-II laser on days 0, 4, 8, 12, 16, and 20, respectively. (b-e) Orthotopic liver tumor luminescence levels of mice after various treatments, including (b) PBS, (c) Nano-BFF, (d) Nano-BFF+NIR-I laser, and (e) Nano-BFF + NIR-II laser. (f) Relative orthotopic liver tumor luminescence levels of mice, and (g) Body weight variations of mice in various treatment groups within 20 days. (h) Photographs of liver tissues in each treatment group, where the orthotopic tumor tissues were highlighted with red dotted circles ($n = 5$). (i) Histological images of the orthotopic liver tumor tissues using H&E, TUNEL, and Ki-67 staining assays (I: PBS; II: Nano-BFF; III: Nano-BFF + NIR-I laser; IV: Nano-BFF + NIR-II laser). Scale bars: 100 μm .

Supplementary Fig. 25. H&E staining images of major organs from various treatment groups, including PBS, Nano-BFF, Nano-BFF + NIR-I laser, and Nano-BFF + NIR-II laser groups. Scale bars: 100 μm .

3. Figure 2I shows that at 4 mm penetration depth, the delta T value for 808 nm is approximately 4 degree less than that for 1064 nm. In Figure 6c however 1064 nm seems to produce 7 degree higher than 808 nm even *in vivo* (Page 9 line 270-271). These results have to be reconciled.

Response: Thank you for this comment. As shown in **Fig. 2a,b,e,f**, the temperature elevations of Nano-BFF were highly dependent on the irradiation time and power density of laser irradiation. The administrated doses of Nano-BFF were 100 $\mu\text{g mL}^{-1}$ and 20 mg kg^{-1} for *in vitro* and *in vivo* studies at 4 mm of penetration depth, respectively. Moreover, the tumor temperature increments of Nano-BFF-treated mice under NIR laser irradiation were relied on the accumulation amounts of Nano-BFF at tumor sites. These factors will lead to a significant difference in temperature increment between *in vitro* and *in vivo* studies under the same penetration depth. In addition, similar results were also observed in previous reports (*ACS Nano* **12**, 2643-2651 (2018); *Adv. Mater.* **30**, 1705980 (2018).).

Furthermore, based on your suggestion, we also reassessed the temperature elevation of Nano-BFF solution under 808 nm or 1064 nm laser irradiation through 4 mm of penetration depth. As shown in **Fig. 2k,l** and **Supplementary 14e,f**, the temperature increment of Nano-BFF solution (100 $\mu\text{g mL}^{-1}$) reached 13.2 $^{\circ}\text{C}$ under 1064 nm laser irradiation, which was 6.4 $^{\circ}\text{C}$ higher than that for 808 nm laser irradiation under the same experimental condition. For *in vivo* studies, the tumor temperature of Nano-BFF (20 mg kg^{-1})-administrated mice reached to 52 $^{\circ}\text{C}$ after 1064 nm laser

irradiation, which was 7 °C higher than that of 45 °C for 808 nm laser irradiation (Fig. 6c). We have updated the related data in the revised manuscript (Fig. 2k,l) and supplementary information (Supplementary Fig. 14e,f).

Supplementary Fig. 14. (e,f) Photothermal conversion curves of Nano-BFF aqueous solution upon exposure to (e) 808 nm and (f) 1064 nm laser irradiation under varied tissue thicknesses (0, 2, 4, 6, 8, and 10 mm).

Fig. 2. (k) Real-time temperature, and (l) temperature variations of Nano-BFF aqueous solution upon the laser irradiation after penetration through tissues with increasing skin thickness.

4. Figure 2a&c vs. e&g appear to indicate that the photothermal efficiency at 808 nm is higher than at 1064 nm: the concentration was the same in both cases, but the temperature was apparently higher at 808 nm than 1064 nm. More details are required to show the calculation for 28.6% for 808 nm and 34.3% for 1064 nm.

Response: Thank you very much for your kind reminder. We have recalculated the photothermal conversion efficiency of Nano-BFF according to the previous method (*J. Am. Chem. Soc.* **139**, 16235-16247 (2017); *Angew. Chem. Int. Ed.* **57**, 3963-3967 (2018)). Specifically, the photothermal conversion efficiency of Nano-BFF was calculated by monitoring the temperature change of Nano-BFF in aqueous dispersion as a function of

time under continuous laser irradiation. When the temperature reached to a plateau, the laser was turned off and the temperature was recorded during the cooling stage until the temperature decreased to the room temperature.

Details are listed as follows:

Based on the total energy balance for this system:

$$\sum_i m_i C_{p,i} \frac{dT}{dt} = Q_s - Q_{loss}$$

where m_i is the mass and $C_{p,i}$ is the heat capacity of system components, respectively. Q_s is the photothermal heat energy input by irradiating Nano-BFF solution with NIR laser, and Q_{loss} is thermal energy lost to the surroundings. When the temperature is maximum, the system is in balance.

$$Q_s = Q_{loss} = hS\Delta T_{max}$$

where h is heat transfer coefficient, S is the surface area of the container, ΔT_{max} is the maximum temperature change.

The photothermal conversion efficiency is calculated from the following equation:

$$\eta = \frac{hS\Delta T_{max}}{I(1-10^{-A\lambda})}$$

where I is the laser power and λ is the absorbance of Nano-BFF solution at the wavelength of 1064 nm or 808 nm.

In order to get the hS , a dimensionless driving force temperature, θ is introduced as follows:

$$\theta = \frac{T - T_{surr}}{T_{max} - T_{surr}}$$

where T is the temperature of Nano-BFF solution, T_{max} is the maximum system temperature, and T_{surr} is the initial temperature

And a sample system time constant τ_s ,

$$\tau_s = \frac{\sum_i m_i C_{p,i}}{hS}$$

Thus

$$\frac{d\theta}{dt} = \frac{1}{\tau_s} \frac{Q_s}{hS\Delta T_{max}} - \frac{\theta}{\tau_s}$$

When the laser is off, $Q_s = 0$, therefore $\frac{d\theta}{dt} = -\frac{\theta}{\tau_s}$, $t = -\tau_s \ln \theta$. So, hS could be calculated from the slope of cooling time vs $\ln \theta$. The time constant (τ_s) of heat transfer from Nano-BFF was determined to be 220.42 s and 185.01 s for 808 nm and 1064 nm, respectively (**Fig. 2c,g**). The ΔT_{max} of Nano-BFF was 25.6 °C and 24.1 °C for 808 nm and 1064 nm laser irradiation, respectively. Therefore, the photothermal conversion efficiency (η) of Nano-BFF was calculated to be 28.6% and 34.3% at 808 nm and 1064

nm, respectively. We have clarified this issue and added the related data in the revised manuscript (**Page 5 and Fig. 2c,g**) and supplementary information (**Page S4**).

5. No biodistribution data of the nanoparticles after iv injection. How many tumor uptakes?

Response: Thank you for this comment. According to your suggestion, we conducted *in vivo* biodistribution analysis to evaluate the accumulation of Nano-BFF in the main organs and tumor tissues at different time points post-injection. **Supplementary Fig. S18** illustrated that the Nano-BFF mainly distributed in the liver tissues of mice. The accumulation of Nano-BFF at tumor sites was relatively high with the amounts of 4.46% ID g⁻¹ at 4 h and 8.57% ID g⁻¹ at 12 h respectively (n = 3), which was ascribed to the typical enhanced permeability and retention effect and prolonged blood circulation time of Nano-BFF. We have clarified this issue and added the related data in the revised manuscript (**Page 8**) and supplementary information (**Supplementary Fig. S18**).

Supplementary Figure S18. *In vivo* biodistribution of Nano-BFF in major organs and tumor tissues (ID% per tissue) after intravenous injection of Nano-BFF at various time points (4, 12, 24, and 48 h) (n = 3).

6. The injection dose was 20 mg/kg for both photo acoustic imaging and photo thermal therapy. This is quite high. Did the authors try lower concentrations?

Response: Thank you for this constructive question. Based on your suggestion, various doses (5 and 10 mg kg⁻¹) of Nano-BFF were administrated into 4T1 tumor-bearing mice, and PA imaging and photothermal therapy studies were further conducted. By utilizing the strong absorbance of Nano-BFF in both NIR-I and NIR-II biowindows, *in vivo* PA imaging analysis was performed to track the time-dependent distribution of Nano-BFF post-injection. Different doses (5, 10, and 20 mg kg⁻¹) of Nano-BFF were intravenously injected into 4T1 tumor-bearing mice, and the PA images and intensities at tumor sites

were recorded at various post-injection time points. As exhibited in **Fig. 4f-h** and **Supplementary Fig. 19**, the PA signal intensity increased gradually over time and reached its maxima at 12 h post-injection, confirming that the increased PA signal was resulted from the efficient accumulation of Nano-BFF.

To evaluate the *in vivo* anti-tumor efficacy of Nano-BFF in the NIR-II biowindow, various doses of Nano-BFF were intravenously injected into 4T1 tumor bearing mice at 12 h post-injection. As showed in **Supplementary Fig. 24a-c,e**, the tumor temperature increased by 14.1 and 17.2 °C at the injection doses of 5 and 10 mg kg⁻¹ under NIR-II laser irradiation respectively, indicating excellent hyperthermia effect from Nano-BFF upon NIR-II laser irradiation. Importantly, Nano-BFF (5 mg kg⁻¹) plus NIR-II laser exposure caused significant tumor suppression with the tumor growth rate of 67.6%. Complete tumor ablation without further recurrence was observed for the mice received with Nano-BFF (10 mg kg⁻¹) administration and NIR-II laser irradiation. In addition, no significant body weight variation was detected in all treatment groups, demonstrating negligible side effect of Nano-BFF towards mouse health (**Supplementary Fig. 24d**). We have clarified this issue and added the related data in the revised manuscript (**Page 8 and Page 10**) and supplementary information (**Supplementary Fig. 19 and 24**) according to your kind suggestion.

Supplementary Fig. S19. (a) *In vivo* PA images of tumors under excitation at 900 nm at different concentrations, and (b) corresponding PA values after intravenous injection of Nano-BFF at various time points.

Supplementary Fig. S24. (a) IR thermal images, and (b) temperature variations at the tumor sites of the mice after different treatments. (c) Relative tumor volumes, (d) body weights of the mice, and (e) tumor growth rates in different treatment groups, including Nano-BFF (5 mg kg⁻¹) + NIR-II laser, and Nano-BFF (10 mg kg⁻¹) + NIR-II laser groups.

7. The proton NMR spectra of formazan and BFF appear to indicate the presence of some impurity in the sample.

Response: Thank you for this comment. According to your suggestion, formazan and BFF were further purified and confirmed by ¹H NMR. The updated ¹H NMR spectra were added in the revised supplementary information (**Supplementary Fig. 1 and Fig. 4**).

Supplementary Fig. 1. ¹H NMR spectrum of formazan (1).

Supplementary Fig. 4. ¹H NMR spectrum of BF₂-formazanate (2).

8. Figure 6C; the green line should be Nano-BFF+NIR-I instead of PBS+NIR-II.

Response: Thank you for this reminder. We apologize for this careless mistake. We have corrected it in the revised manuscript (**Fig. 6C**) and double checked the manuscript carefully.

Fig. 6. (c) Temperature profiles at tumor regions of tumor-bearing mice in different treatment groups after penetration through 4 mm of chicken breast tissue.

9. No statistical analysis in the work. MPE should be calculated in the phototherapy experiments.

Response: Thank you for your reminder. We have added the statistical significance analysis into the revised manuscript (**Page 15; Fig. 5c,d,f,g, Fig. 6d,e, and Fig. 7f**) and supplementary information (**Supplementary Fig. 15b, Fig. 22b, Fig. 23a,c, and Fig. 24c,e**). All data were presented as mean±standard deviation unless otherwise stated. The statistical significance was determined using a two-sided Student's test (* $P < 0.05$, ** $P < 0.01$, and *** $P < 0.001$) unless otherwise stated.

In order to assess the deep-tissue penetration capability of both NIR-I and NIR-II lasers, the power density of 808 nm or 1064 nm laser after penetration into the chicken breast tissue with increasing thickness (2, 4, 6, 8, and 10 mm) was monitored by an optical power meter (**Fig. 2i,j**). In comparison with 1064 nm laser, the residue power density of 808 nm laser attenuated more rapidly at each tissue depth. For instance, after the penetration through 2 mm of chicken breast tissue, the residue power density for 1064 nm and 808 nm lasers was determined to be 0.9 W cm^{-2} and 0.6 W cm^{-2} under the same original power density, respectively. This difference was contributed to better transmittance capability and higher maximum permission exposure (MPE) of 1064 nm laser (1 W cm^{-2}) in comparison with that of 808 nm laser (0.33 W cm^{-2})⁴⁵⁻⁴⁷. We have clarified this issue in the revised manuscript (**Page 6**) according to your suggestion.

References:

45. Y. Liu, W. Zhen, Y. Wang, J. Liu, L. Jin, T. Zhang, S. Zhang, Y. Zhao, S. Song, C. Li, J. Zhu, Y. Yang, H. Zhang, *Angew. Chem. Int. Ed.* **58**, 2407-2412 (2019).
46. M. Ma, N. Gao, X. Li, Z. Liu, Z. Pi, X. Du, J. Ren, X. Qu, *ACS Nano* **14**, 9894-9903 (2020).
47. B. Guo, Z. Sheng, D. Hu, C. Liu, H. Zheng, B. Liu, *Adv. Mater.* **30**, 1802591 (2018).

REVIEWERS' COMMENTS

Reviewer #1 (Remarks to the Author):

Since authors have completely addressed all the critical issues, I recommend the acceptance of this manuscript at current edition.

Reviewer #2 (Remarks to the Author):

The authors made a strong effort in revising the manuscript and addressing my prior comments. The work has been improved significantly, and may be accepted for publication now.

Point-by-point response to reviewers' comments

Comments and suggestions from reviewer 1#:

Since authors have completely addressed all the critical issues, I recommend the acceptance of this manuscript at current edition.

Response: Thanks very much for your recommendation of our work.

Comments and suggestions from reviewer 2#:

The authors made a strong effort in revising the manuscript and addressing my prior comments. The work has been improved significantly, and may be accepted for publication now.

Response: Thanks very much for your recommendation of our work.